# IoT-Assisted Automatic Driver Drowsiness Detection through Facial Movement Analysis Using Deep Learning and a U-Net-Based Architecture

Shiplu Das [1] , Sanjoy Pratihar [1] , Buddhadeb Pradhan [2], Rutvij H. Jhaveri [3] and Francesco Benedetto [4],*

1   Computer Science and Engineering, Indian Institute of Information Technology, Kalyani 741235, India; shiplu_phd21@iiitkalyani.ac.in (S.D.); sanjoy@iiitkalyani.ac.in (S.P.)
2   Computer Science and Engineering, University of Engineering and Management, Kolkata 700160, India; buddhadeb.pradhan@uem.edu.in
3   Computer Science and Engineering, School of Technology, Pandit Deendayal Energy University, Gandhinagar 382007, India; rutvij.jhaveri@sot.pdpu.ac.in
4   Signal Processing for Telecommunications and Economics, Roma Tre University, 00154 Roma, Italy
*   Correspondence: francesco.benedetto@uniroma3.it

**Abstract:** The main purpose of a detection system is to ascertain the state of an individual's eyes, whether they are open and alert or closed, and then alert them to their level of fatigue. As a result of this, they will refrain from approaching an accident site. In addition, it would be advantageous for people to be promptly alerted in real time before the occurrence of any calamitous events affecting multiple people. The implementation of Internet-of-Things (IoT) technology in driver action recognition has become imperative due to the ongoing advancements in Artificial Intelligence (AI) and deep learning (DL) within Advanced Driver Assistance Systems (ADAS), which are significantly transforming the driving encounter. This work presents a deep learning model that utilizes a CNN–Long Short-Term Memory network to detect driver sleepiness. We employ different algorithms on datasets such as EM-CNN, VGG-16, GoogLeNet, AlexNet, ResNet50, and CNN-LSTM. The aforementioned algorithms are used for classification, and it is evident that the CNN-LSTM algorithm exhibits superior accuracy compared to alternative deep learning algorithms. The model is provided with video clips of a certain period, and it distinguishes the clip by analyzing the sequence of motions exhibited by the driver in the video. The key objective of this work is to promote road safety by notifying drivers when they exhibit signs of drowsiness, minimizing the probability of accidents caused by fatigue-related disorders. It would help in developing an ADAS that is capable of detecting and addressing driver tiredness proactively. This work intends to limit the potential dangers associated with drowsy driving, hence promoting enhanced road safety and a decrease in accidents caused by fatigue-related variables. This work aims to achieve high efficacy while maintaining a non-intrusive nature. This work endeavors to offer a non-intrusive solution that may be seamlessly integrated into current automobiles, hence enhancing accessibility to a broader spectrum of drivers through the utilization of facial movement analysis employing CNN-LSTM and a U-Net-based architecture.

**Keywords:** artificial intelligence; advanced driver assistant systems; Internet of Things; U-Net; automated vehicles; convolutional neural network–long short-term memory

## 1. Introduction

In recent years, the rapid advancement of technology has led to the convergence of various fields, giving rise to innovative solutions that enhance safety, efficiency, and convenience in everyday life. One such intersection is the realm of IoT and automotive safety, where the application of IoT principles to automobiles has paved the way for groundbreaking advancements in driver assistance systems [1]. Among these innovations, the detection of driver drowsiness holds immense significance in preventing accidents

and ensuring road safety. The IoT-assisted automatic driver drowsiness detection system offers a multi-tiered approach to road safety. Firstly, it operates in real time, providing instantaneous feedback to the driver and triggering alerts if drowsiness is detected. These alerts can take various forms, such as auditory alarms, haptic feedback, or even in-vehicle adjustments to lighting and climate control. Secondly, the system contributes to data-driven insights by collecting and analyzing a wealth of information over time. These data can be utilized for statistical analysis, identifying trends, and refining algorithms to enhance detection accuracy.

A vehicle is the most powerful thing on the road. When used recklessly, it may be dangerous, and occasionally, the lives of other road users may be at risk. Failure to realize when we are too tired to drive is a type of carelessness. Many academics have authored study papers on driver tiredness detection systems to monitor and prevent a disastrous outcome from such recklessness. As a result, this work was carried out to provide a new viewpoint on the current situation and optimize the solution. The most common cause of accidents at night is driving when tired. Fatigue and drowsiness are regularly to blame for serious accidents on roadways [2,3]. Only identifying tiredness and warning the driver will solve this issue. Drivers are more prone to falling asleep on trips requiring long stretches of driving on regular routes such as highways. High-risk trips are those made for work-related purposes, particularly those involving truck drivers. Another category of high-risk travel [4,5] involves corporate car drivers. As a result, there is a clear relationship between the time of day and the likelihood of falling asleep behind the wheel. These observations highlight the interplay between driving circumstances, specific driver profiles, and the temporal aspect of the risk of drowsy driving. Recognizing these correlations is essential for developing effective strategies to mitigate the dangers associated with driver drowsiness and enhance overall road safety.

The public health issue of motor vehicle collisions (MVCs) and injuries is global [6]. Drivers' sleepiness and weariness are major contributing factors to fatal crashes and MVC risk factors. The research is well-referenced about the prevalence of driver fatigue, drowsiness, and weariness, as well as its effects on the incidence of MVCs and injuries from traffic accidents. The pattern of acute weariness, exhaustion, persistent sleepiness, sleep issues, and high workload has been connected to poor performance in psychomotor tests and driving simulators due to the rising incidence of MVCs, injuries, and deaths in specific populations. A widely used instrument for gauging self-reported driving behavior and finding a connection between it and accident involvement is the Driver Behaviour Questionnaire (DBQ) [7]. As human mistakes cause most traffic accidents, the DBQ is one of the most often-used research instruments to explain erroneous driving behaviors in three basic categories, including errors, infractions, and lapses. The authors then suggested incorporating the study's findings into micro-simulations to more precisely imitate drivers' actions on urban street networks. Drowsy driving impacts everyone, regardless of age, profession, economic situation, or level of driving expertise. Drivers frequently feel tired, and there are occasions when people have to drive while being severely sleep-deprived. Teenagers and new drivers have spent less time on the road. Thus, their driving skills have not yet matured. Younger drivers are also more inclined to drive after hours for social or professional reasons, which increases their risk of driving while fatigued. Shift and night workers frequently put in long hours at the office and are usually worn out when it is time to clock out. They do not have a long journey home, yet many still try to use their cars out of habit and duty. The risk of sleepy driving is six times higher for people working nights, rotating, or double shifts compared to other categories of workers. Doctors, nurses, pilots, police officers, and firefighters are just a few occupations that frequently have long shifts. Compared to the typical commuter, people who drive for a living log more kilometers on the road. Because many commercial drivers work long hours and face strict deadlines, they also have a considerably higher risk of driving while fatigued. Regular business travelers are especially vulnerable to the dangers of drowsy driving because they frequently experience jet lag and switch time zones as frequently as they do ZIP codes. Getting enough

sleep cannot be easy if people travel a lot for work, making it challenging to stay safe on the road. For drivers with sleep disorders, drowsy driving can be a daily struggle. Some drivers may experience daytime exhaustion and drowsiness due to narcolepsy or insomnia, but those with untreated obstructive sleep apnea (OSA) are at a significantly higher risk of experiencing these issues. Some drugs can also have the opposite effect, causing sleepiness in drivers when they need to be focused behind the wheel.

Sleep-related crashes are more likely to result in catastrophic injuries, possibly due to the higher speeds involved and the driver's incapacity to avoid an accident or even stop in time [8]. Drowsiness can be understood in many ways, like the tendency to yawn, sleepiness, tiredness, and others. This causes a significant number of fatal accidents and deaths. It is currently a hot topic for research. In summary, this paper seeks to advance road safety by detecting driver drowsiness and issuing timely alerts, thereby reducing the risk of accidents linked to fatigue. This research leverages the IoT and deep learning technologies to create a system that is not only effective but also unobtrusive, making use of facial movement analysis to improve driver safety on the road. Ultimately, the goal is to make this solution easily accessible to a wider range of drivers, thereby contributing to safer roadways and a reduction in accidents caused by drowsy driving. Section 1 presents an introduction to this paper. Section 2 presents the contributions of the paper. Section 3 presents the related works on various methods of drowsiness detection using different machine learning and deep learning techniques. Section 4 designs the architecture and mathematical analysis of the proposed model. Section 5 describes the analysis and discussion of the results. The final section summarizes our research findings and future plans.

## 2. Contributions of this Paper

By addressing a critical safety concern on the road, this study can help reduce accidents caused by drowsy driving. This study can substantially impact road safety by combining the IoT, deep learning, and facial movement analysis to automatically detect driver drowsiness. The contributions of this paper are as follows:

- This paper presents U-Net-based segmentation, which only takes information from the physical regions of the driver's body. After segmentation, we encode the image information and combine data from multiple time steps. Then, we minimize the effect of an external factor, and U-Net-based segmentation is carried out before passing the frames to the model.
- After that, the segmented body region is fed to the CNN-LSTM model, which generates a softmax output, indicating the probability of the driver being drowsy.
- The method combines segmentation, image feature extraction, and time-series analysis algorithms to confidently make the classification decision.
- This paper leverages IoT principles to develop a real-time monitoring system. By strategically placing sensors within the vehicle and utilizing interconnected data transmission, we pioneer a practical application of IoT technology in the context of driver safety.
- This paper identifies and addresses challenges associated with accurate drowsiness detection, such as minimizing false positives and negatives and accommodating various driving scenarios.

## 3. Related Works

Driving while tired increases the likelihood of a collision or accident. Many individuals are killed in automobile accidents yearly due to driving caused by a lack of sleep, drug and alcohol abuse, or heat exposure. Accurate drowsiness detection based on eye state has been achieved using a variety of indicators and parameters, as well as the expertise of specialists. An essential component of sleepiness detection is predicting facial landmarks, detecting eye states, and presenting the driver's status on a screen. Major traffic accidents frequently occur when the driver feels tired from long hours of driving, a physical sickness, or even alcohol. Drowsiness can be defined as a natural state where an individual feels

exhausted. The individual's reflex is significantly reduced, which can cause the driver to be unable to take quick actions when necessary. Also, studies have shown that driving performance worsens with increased drowsiness. A human can quickly tell if someone is tired by detecting specific actions or behaviors. Drowsy driving is a serious issue that affects the driver, puts other people's lives in danger, and harms the nation's infrastructure. There has been an enormous surge in the daily use of private transportation in modern society. When traveling a long distance for an extended period, driving can become monotonous. Traveling for a long time without any rest or sleep is one of the key reasons drivers lose focus.

Detection methods follow eye movements and facial expressions to identify the drowsiness state of the driver with the help of convolutional neural networks (CNN). Recently, convolutional neural networks (CNN) have also been used in a method that helps in behavioral recognition by understanding upper body postures and producing the image's corresponding state as output. Using that proposed recognition model, some data were collected related to driving. Another proposed model detects whether a person is busy with phone calls and one hand is on the steering wheel. The method is based on the Faster-RCNN mechanism. Another model is based on an attention mechanism and is different from CNN-based methods. The attention mechanism-based model classifies fine-grained images. However, these mechanisms do not help predict the driver's drowsiness while driving and do not focus on the distracting scenes inside the vehicle. Therefore, it is difficult to identify the driver's actions while moving. Another proposed model identifies the position of faces through various poses. Existing methods for detecting driver drowsiness can be categorized into three kinds: physiological, vehicle-based, and behavioral.

The first type of method attaches a device to the driver's skin. In [9], Awais et al. exploited the use of ECG and EEG characteristics. First, EEG features are collected, such as time-domain statistical descriptors, complexity metrics, and power spectrum measures, as well as ECG features, including heart rate, HRV, LF/HF ratio, and other variables. Next, all of these features are combined using SVM, and discrimination is achieved by utilizing these hybrid features.

Another method by Warwick et al. [10] used the idea of a bio-harness. The system works in two phases. The driver's physiological data are gathered in the first phase using a bio-harness. An algorithm analyzes the readings in the second phase. The problem with the methods in this category is that a device has to be attached to the driver's skin, which may only be comfortable for some people. The second type of method analyzes the usage pattern of the vehicle control system, like steering wheel movements, braking habits, and lane departure measurements. These methods use these data to detect driver drowsiness.

Zhenha et al. [11] suggested steering wheel motions over time using a temporal detection window as the primary detection feature. This window is used to evaluate the steering wheel's angular velocity in the time-series analysis by comparing it to the statistical properties of the movement pattern below the fatigue threshold.

Li et al. [12] used data on Steering Wheel Angles (SWAs) to monitor driver drowsiness under natural conditions. The problem with vehicle-based methods is that they are unreliable, which may result in many false positives, thereby significantly affecting the assessment of roads and drivers' driving skills. The third type of method is more reliable compared to the second type, as it only focuses on the driver.

The method proposed by Saradadev et al. [13] used the mouth and yawning as detection features. First, it locates and tracks the mouth using a cascade of classifiers, and then an SVM model is used to analyze and classify a drowsy driver.

Another method by Teyeb et al. [14] analyzed the closing of the eyes and head posture for discrimination. First, the face is partitioned into three regions. Then the wavelet network is used to determine the state of the eyes.

The authors of [15] proposed video-based driver sleepiness detection using real-time techniques, achieving 91.7% accuracy and representing the Karolinska sleepiness scale.

They compared their model to the PERCLOS-based baseline detection method. Figure 1 depicts the various drowsiness detection techniques.

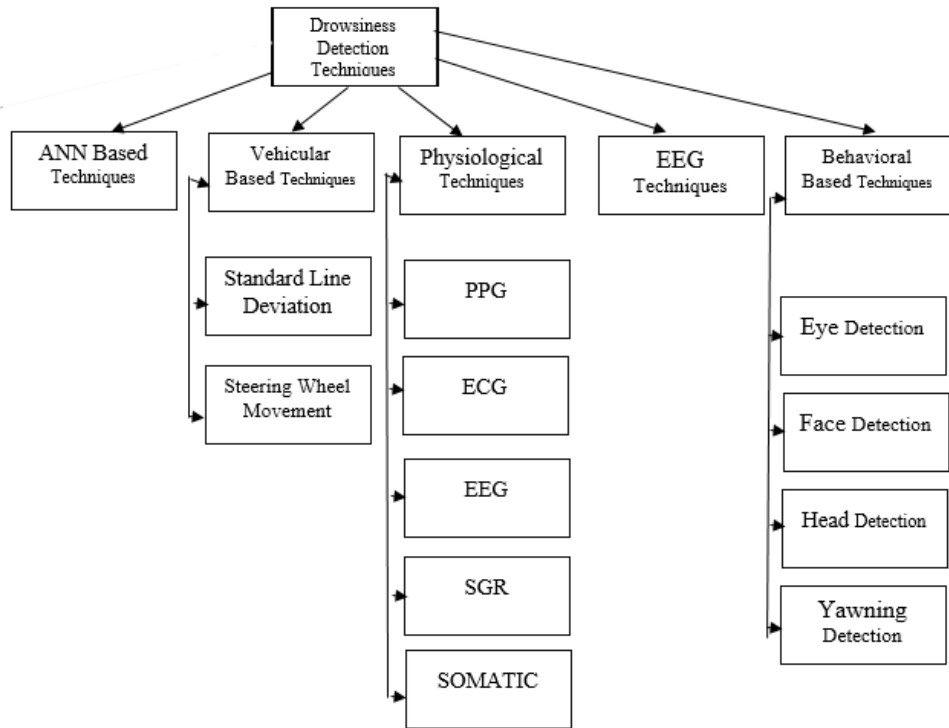

**Figure 1.** Various drowsiness detection techniques.

Alam et al. [16] proposed a deep learning technique based on a convolutional neural network for drowsy driver detection using a signal-channel EEG signal.

In [17], the authors proposed an EEG classification system for driver drowsiness based on deep learning techniques, achieving 90.42% accuracy. They designed two procedures: data acquisition and model analysis. Slow eye closure is often a reliable method for detecting drowsiness, which can be captured by measuring the PERCLOS, i.e., the percentage of eye closure. Issues like different lighting conditions or orientations can influence the system. Upon occurrences of these conditions, the system is blind. The automatic detection of driver fatigue using EEG signals and deep neural networks is a multidisciplinary effort, combining expertise in neuroscience, signal processing, and machine learning. It has the potential to significantly contribute to road safety by preventing accidents caused by drowsy drivers. Sobhan et al. [18] presented a mechanism designed to identify driver fatigue, a critical factor in mitigating traffic accidents. The approach involved collecting information from 11 individuals, resulting in a comprehensive dataset adhering to established standards. The study's findings indicate that the proposed deep CNN-LSTM network demonstrated the ability to hierarchically learn features from the raw EEG data, surpassing the accuracy rates achieved by previous comparison methods in the two-stage classification of driver fatigue.

In [19], Dua et al. proposed the use of different deep learning models, like Alexnet, VGG-Facenet, FlowimageNet, and ResNet, with the softmax classifier, achieving 85% accuracy. In [20], Jamshidi et al. proposed hierarchical deep drowsiness detection, achieving 87.19% accuracy, and used an LSTM network for the temporal information between the frames. The authors proposed a hybrid learning technique with NTHU-DDD and UTA-RLDD.

Liu et al. [21] highlighted the essential use of deep neural networks in such a model within the machine learning field. It has high demand and great value. Hussein et al. [22] presented a study that uses three deep learning-based algorithms—a deep neural network, recurrent neural network, and CNN—to categorize captured driving data using a standard identification procedure and choose the best one for a proposed detection mechanism.

Several approaches were employed to avoid overfitting. The CNN outperformed the other two classification algorithms, with an accuracy of 96.1%, and was thus suggested for the recognition system.

The algorithm proposed in [23] works in two stages. First, it locates and crops the mouth region. Then, in the second stage, an SVM is used to classify whether the image indicates driver fatigue and alerts the driver accordingly. The system uses a cascade of classifiers to locate the mouth region. The SVM is trained on pictures of mouth regions representing various styles of yawning. In contrast, our proposed method uses significantly more information as the input, not just the mouth region. Based on the aforementioned research study, the motive is to control the rate of accident cases due to fatigue or driver drowsiness so that no mishaps occur and, most importantly, to enhance safety in terms of traffic rules and regulations whenever reckless driving takes place due to an unconscious state of mind, i.e., drowsiness. The neural network is trained using the PERCLOS and POM drowsiness thresholds [24].

MT-CNN extracts the face and the feature points, which aid in obtaining the shape of the eyes and mouth. EM-CNN takes action by assessing the conditions of the eyes and mouth. When a threshold is met or surpassed, the degrees of eye and mouth closure are determined by observing the unbroken picture frames; these segmented images of the driver are then passed through blocks of convolutional layers followed by a $1 \times 1$ Conv. Coated for dimension reduction, the output is passed through an LSTM layer. Zhang et al. [25] proposed the use of the AdaBoost, LBF, and PERCLOS algorithms, and the accuracy of the model was 95.18%. The hardware and software required for this method are relatively inexpensive, making it a feasible solution for mass deployment. In a study by Ulrich, L et al. [26], 11 participants participated in an auditory and haptic ADAS experiment while having their attention tracked while driving. The drivers' faces were captured using an RGB-D camera. These pictures were then examined through the use of a deep learning technique, which involved training a convolutional neural network (CNN) designed expressly for facial expression recognition (FER). Studies have been conducted to evaluate potential connections between these findings and ADAS activations, as well as event occurrences or accidents. Different algorithms for driver drowsiness detection are given below in Table 1. Table 2 presents the research gaps in the existing algorithms for drowsiness detection.

**Table 1.** Different algorithms for driver drowsiness detection

| Paper | Algorithms | Accuracy | Advantages | Disadvantages |
|---|---|---|---|---|
| Li et al. [27] | SVM (Support Vector Machine) | 91.92% | Can be integrated with other driver assistance systems | Requires training to interpret EEG data and May be affected by other factors, such as stress or fatigue. |
| Pauly et al. [28] | Histogram of oriented gradient and Support Vector Machine | 91.6% | This method can detect drowsiness in real time, so it can provide early warning signs to the driver. | The SVM classifier needs to be trained on a dataset of images of drowsy and non-drowsy drivers in order to be effective. |
| Flores et al. [29] | Viola–Jones object detection, Adaboost algorithm, neural networks, and Support Vector Machine | - | This system only requires a camera to detect drowsiness, so it is non-intrusive for the driver. | This system may be affected by other factors, such as driver distraction or fatigue. |
| B.Manu et al. [30] | Viola–Jones algorithm, K-means algorithm, SVM | 94.58% | This method is accurate in detecting drowsiness, even in challenging conditions. | This method may be affected by other factors, such as driver distraction or fatigue, and environmental factors. |

**Table 1.** *Cont.*

| Paper | Algorithms | Accuracy | Advantages | Disadvantages |
|---|---|---|---|---|
| Rahman et al. [31] | Viola–Jones algorithm, Adaboost, Haar classifier | 94% | Eye-blink monitoring has the potential to reduce the number of accidents caused by driver drowsiness. | Eye-blink monitoring may be affected by other factors, such as driver distraction or fatigue. |
| Anjali et al. [32] | Viola–Jones object detection, Haar cascaded classifier | - | This strategy has the potential to minimize the number of accidents caused by driver tiredness. | The system needs to be trained on a dataset of eye-blink data from drowsy and non-drowsy drivers in order to be effective. |
| Coetzer et al. [33] | Artificial neural networks, Support Vector Machines, adaptive boosting (AdaBoost) | 98.1% | Challenging conditions such as low lighting and different head poses. | Eye detection may be affected by environmental factors such as lighting and occlusion. |
| Punitha et al. [34] | Viola–Jones, Face Cascade of Classifiers, Support Vector Machine | 93.5% | Eye-state analysis has been shown to be accurate in detecting drowsiness | Ambient elements such as illumination and occlusion may have an impact on eye-state analyses. |

**Table 2.** Research gaps in the different algorithms for driver drowsiness detection.

| Paper | Approach | Key Contribution | Research Gap |
|---|---|---|---|
| Mungra et al. (2020) [35] | CNN-based emotion recognition | High accuracy in detecting fear, anger, and sadness expressions. | Limited investigation on the impact of different CNN architectures and data augmentation techniques. |
| Weng et al. (2022) [36] | Multimodal emotion recognition | Improved accuracy through the multimodal fusion of facial expressions and signals. | Lack of focus on temporal analysis and integration of deep learning architectures like LSTM. |
| Lea et al. (2017) [37] | Temporal convolutional networks | Real-time emotion recognition with accurate fear, anger, and sadness detection. | Limited exploration of combining CNN and LSTM for improved emotion detection. |
| Li et al. (2019) [38] | LSTM-based facial expression recognition | Consideration of temporal context for improved emotion detection. | Absence of spatial analysis and utilization of U-Net architecture for accurate facial feature extraction. |
| Li et al. (2020) [39] | Attention mechanism and CNN | Enhanced discriminative power through an attention mechanism. | Insufficient exploration of combining attention mechanisms with LSTM. |
| Anand et al. (2019) [40] | U-Net architecture for facial analysis | Precise facial feature extraction and localization. | Limited investigation on temporal dynamics and utilization of LSTM for improved emotion detection. |
| Wang et al. (2015) [41] | Facial expression recognition in vehicles | Robust emotion detection addressing the challenges of occlusions and partial views. | Lack of exploration of multimodal fusion and comprehensive temporal analysis for improved accuracy. |

The proposed method's segmentation algorithm, U-Net [42], is simply a collection of convolution and ReLU blocks with some max-pooling layers between the first and second halves, followed by some transpose convolution layers. U-Net is characterized by its U-shaped architecture, which consists of a contracting path (encoder) followed by an expanding path (decoder). This unique design enables it to capture both high-level contexts and fine-grained details in an image. Drowsy face detection often requires analyzing facial features at multiple scales, as signs of drowsiness can manifest differently in different

parts of the face. U-Net's encoder–decoder structure and skip connections enable the network to extract features at various levels of granularity, allowing it to recognize drowsy faces with diverse characteristics. U-Net's ability to handle inputs of varying sizes and adapt to different lighting conditions, poses, and backgrounds makes it robust to real-world image variability. Drowsy face detection systems often need to work in diverse environments, and U-Net's flexibility can help maintain performance across these settings. U-Net typically converges quickly during training, which is beneficial for training drowsy face detection models. Rapid training can save time and computational resources, making it easier to experiment with different model architectures and training data variations. U-Net's efficiency in terms of both training and inference makes it suitable for real-time applications, such as drowsy driver detection systems. This ensures timely warnings or interventions when drowsiness is detected. The ability of U-Net to capture subtle facial cues and contexts can help reduce false positives in drowsy face detection. This ensures that alarms are triggered only when genuine signs of drowsiness are present, enhancing the user experience and avoiding unnecessary interruptions. U-Net is a powerful and versatile architecture for drowsy face detection in image-to-image mapping tasks. Its ability to capture spatial information, extract multi-scale features, and adapt to varying conditions contributes to the accuracy and reliability of drowsy face detection systems, making them valuable for driver safety and other applications where monitoring facial expressions is critical. Finally, in terms of operation, the transposed convolution multiplies the filter value by the encoded matrix to produce another padded matrix with a more excellent resolution. This stage displays the output of the LSTM layer, which can be combined with another system to create a functional end-to-end system. For example, if a buzzer is linked at the end and the driver is identified as tired, the buzzer will sound, or in a self-driving car, the car will safely stop on the side of the road and then do something to wake the user up. D Gao et al. [43] described federated learning based on Connection Temporal Classification (CTC) for the heterogeneous IoT. Federated learning involves training machine learning models across decentralized devices while keeping data on the devices, addressing privacy and communication challenges. The authors proposed FLCTC, a federated learning system based on CTC for heterogeneous IoT applications, and tested the system in forest fire predictions to illustrate its applicability. This integration enhanced the capabilities of both the IoT and ML, enabling intelligent decision making, automation, and insights from the the vast amounts of data generated by IoT devices.

Temporal analysis is crucial in various applications, and integrating deep learning architectures like LSTM (Long Short-Term Memory) can indeed enhance the ability to model temporal dependencies in data. LSTMs are particularly effective in handling sequences and time-series data due to their ability to capture long-range dependencies. The combination of convolutional neural networks (CNNs) and Long Short-Term Memory (LSTM) networks is a powerful approach, especially for tasks like emotion detection. CNNs are excellent at extracting spatial features from data, whereas LSTMs excel at capturing temporal dependencies. In the context of emotion detection, a common approach is to use CNNs to extract relevant features from input data (such as images or sequences of frames), and then feed these features into an LSTM for capturing temporal dynamics. Combining attention mechanisms with LSTM is a fantastic avenue for improving the performance of models dealing with sequential data. Attention mechanisms enable the model to focus on specific parts of the input sequence, making it more adaptable and effective in capturing relevant information. The "DistB-SDCloud" architecture, which improves cloud security for intelligent IIoT applications, was presented in [44]. In order to maintain flexibility and scalability while offering security, secrecy, privacy, and integrity, the suggested architecture employs a distributed BC technique. Clients in the industrial sector profit from BC's efficient, decentralized, and distributed environment. The paper also presented an SDN technique to enhance the cloud infrastructure's resilience, stability, and load balancing.

The authors of [45] proposed a lightweight and robust authentication system for WMSN, which integrates physically unclonable functions (PUFs) and state-of-the-art blockchain technology, to address these two major concerns. Furthermore, a fuzzy extractor approach was presented to handle biometric data. Two security evaluation techniques were then employed to demonstrate the excellent reliability of the suggested approach. Lastly, among the compared systems, the suggested mutual authentication protocol required the lowest computing and communication costs, as demonstrated in performance evaluation trials.

Zhou et al. [46] presented the domains of two input vehicle images that were transformed into other domains in the network structure using a generative adversarial network (GAN)-based domain transformer. A four-branch Siamese network was then created to learn the two distance metrics between the images in the two domains. In order to calculate the ultimate similarity between the two input photos for vehicle Re-ID, the two distances were finally merged. The outcomes of the experiments indicated that the suggested GAN-Siamese network architecture attained cutting-edge results on four extensive vehicle datasets: VehicleID, VERI-Wild, VERI-Wild 2.0, and VeRi776. Zhou, Z et al. [47] identified boundary frames as possible accident frames based on the generated frame clusters. Next, in order to verify whether these frames were indeed accident frames, the authors recorded and encoded the spatial relationships of the items identified from these potentially accident frames. Comprehensive tests showed that the suggested method satisfied the real-time detection requirement in the VANET environment and provided promising detection efficiency and accuracy for traffic accident detection. Zhou, Z et al. [48] introduced a novel identity-based authentication system. The proposed method demonstrated secure communication between various components of the green transport system through the use of lightweight authentication mechanisms. Zhou, Z et al. proposed HAR [46], a robust subspace-clustering (SOAC-RSC) scheme based on sequential order-aware coding. Two expressive coding matrices are learned in a sequential order-aware manner from unconstrained and restricted films, respectively, by feeding the motion properties of video frames into multi-layer neural networks to generate the appropriate affinity graphs.

Khajehali et al. [49] presented a complete systematic literature review focusing on client selection difficulties in the context of federated learning. The goal of this SLR was to support future CS research and development in FL. Deng et al. [50] presented an iterative optimization approach for EE under conditions involving interference constraints and minimal feasible rates for secondary users. In the first step, Dinkelbach method-based fractional programming is used with a given UAV trajectory to determine the appropriate gearbox power factors. In the second step, the successive convex optimization technique is used to update the system parameters using the prior power allocation scheme. Finally, to find the optimal UAV trajectory, reinforcement learning-based optimization is used. Sarkar et al. [51] suggested that the Industrial Internet of Things (IIoT) has gained importance at a time when the medical industry's potential is rapidly rising. To address this, the authors presented the Intelligent Software-defined Fog Architecture (i-Health). Based on each patient's past data patterns, the controller decides whether to transport data to the fog layer.

The fusion of IoT technology and facial movement analysis has led to the creation of an innovative solution for enhancing driver safety. By harnessing real-time data acquisition and advanced machine learning techniques, the IoT-assisted automatic driver drowsiness detection system has the potential to significantly reduce accidents caused by driver fatigue. As technology continues to evolve, this system is a testament to the power of interdisciplinary collaboration in creating impact solutions that can shape the future of road safety. Here, our contribution extends to the novel data acquisition methodology by capturing a range of facial movements and expressions, including eye-closure duration, blinking patterns, and head orientation. In this way, we are able to acquire real-time data crucial for accurate drowsiness detection.

## 4. Proposed Model

Drowsiness is a big issue while driving; therefore, some drowsiness detection solutions must be implemented in front of a driver while they are driving a vehicle. So, with the help of the OpenCV and Dlib libraries, we developed a driver drowsiness detection system that initiates whether the person's eyes are closed or open, i.e., the eyes are in an active state or a passive (lazy) state. Moreover, the main motive is to identify or detect whether the person is yawning while holding the steering wheel. It becomes essential to implement such a detection system to reduce accidents caused by fatigue resulting from tiredness or sleepiness, which is more dangerous at night, with accident cases increasing by more than 50 percent. So, to reduce the number of road accidents, an advanced method of detection must be able to be implemented in real-world scenarios. The motive of this study is to control the rate of accidents due to fatigue or driver drowsiness so that no mishaps occur and, most notably, to enhance safety in terms of traffic rules and regulations whenever reckless driving takes place due to an unconscious state of mind, i.e., drowsiness. The proposed detection method follows the physical nature, i.e., eye movement and facial expression, to identify the drowsy state of the driver with the help of convolutional neural networks (CNN). The proposed model uses a 15 s video clip as input. The video is sampled at 1 s intervals, yielding 15 frames. These frames are then passed through a U-Net to extract the region of interest (ROI), in this case, the driver's body. Using 1 × 1 Conv layers significantly reduces the dimension of the output we obtain from the convolution layers, which plays a significant role in encoding the features extracted from the input frames.

U-Net has been a very successful model when it comes to image-to-image mapping. Detecting faces is a very intensive task that can be challenging in real-world situations due to variations in driver posture and environmental factors such as radiance or occlusions. Using the depth-cascading multitasking framework, we can more easily align and detect faces, improving internal relations through facial features like the positions and locations of the right and left eyes, corners of the mouth, and nose. From the architecture of multitask cascaded convolutional networks, we can understand the comparisons between the P (Proposal)-, R (Refined)-, and O (Output)-Nets. These three sub-networks detect the face and feature points. In the P-Net, different-sized image pyramids are assembled in a sequence as input. A convolutional network determines whether a 12 × 12 face exists at each position. Then, a boundary box is calibrated with a regression vector to remove overlapping face regions. Figure 2 represents the architecture of the P-Net, whereas Figure 3 shows a 24 × 24 reshaped R-Net image. Boundary box regression and non-maximum value suppression shield the face window. A connection layer is added to the network structure to acquire an accurate face position. The O-Net image is reshaped to 48 × 48 to output the final position of the face along with the facial feature points. To unify all real-world images of different sizes, a convolution layer resizes them to 175 × 175, and pooling also acquires a 44 × 44 × 56-sized feature map.

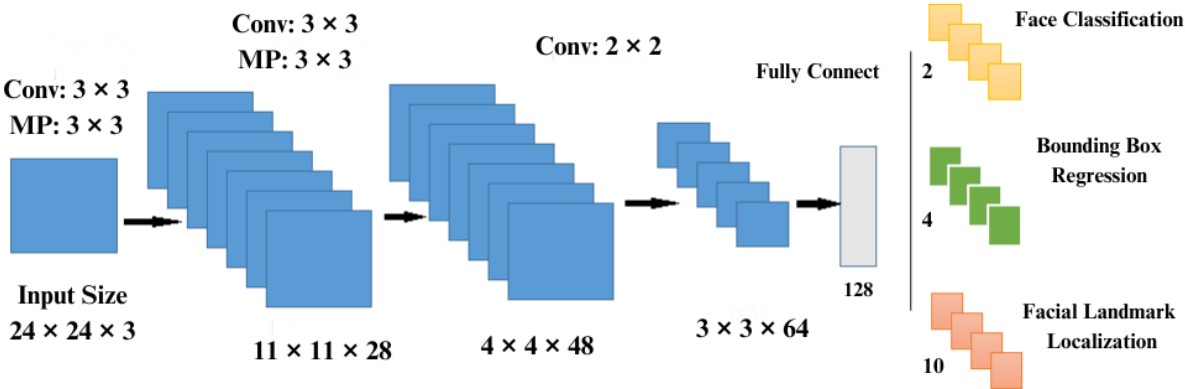

**Figure 2.** P-Net architecture.

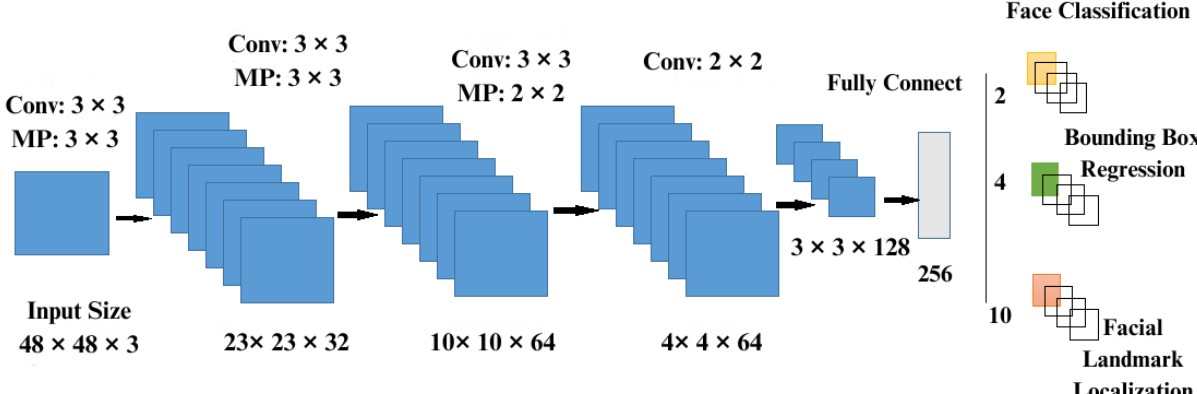

**Figure 3.** R-Net architecture.

The convolution layer utilizes a 3 × 3 convolution kernel with a step size of 1, whereas the pooling layer adopts a 3 × 3 configuration with a step size of 2. A pixel layer is used to prevent size reduction, which causes a loss of details at the borders. Now, three pooling layers increase adaptability through 3 × 3 pooling with sizes of 1 × 1, 3 × 3, and 5 × 5. Another resulting pooling map is a 44 × 44 × 256 feature map. Then, an 11 × 11 × 72 feature map is generated by channeling through three layers of the residual block. A one-dimensional vector is created from the feature map and linked layer to reduce the parameters through random inactivation, minimizing overfitting. Using softmax, we can now define the eyes and mouth as open or closed. Although there is a similar network for time-series data (GRU), Long Short-Term Memory (LSTM) is better at retaining information longer, which helps associate specific patterns when embedding the frames from the video clip. Lastly, the final block is composed of fully connected layers followed by softmax activation. Figure 4 depicts our proposed U-Net-based architecture.

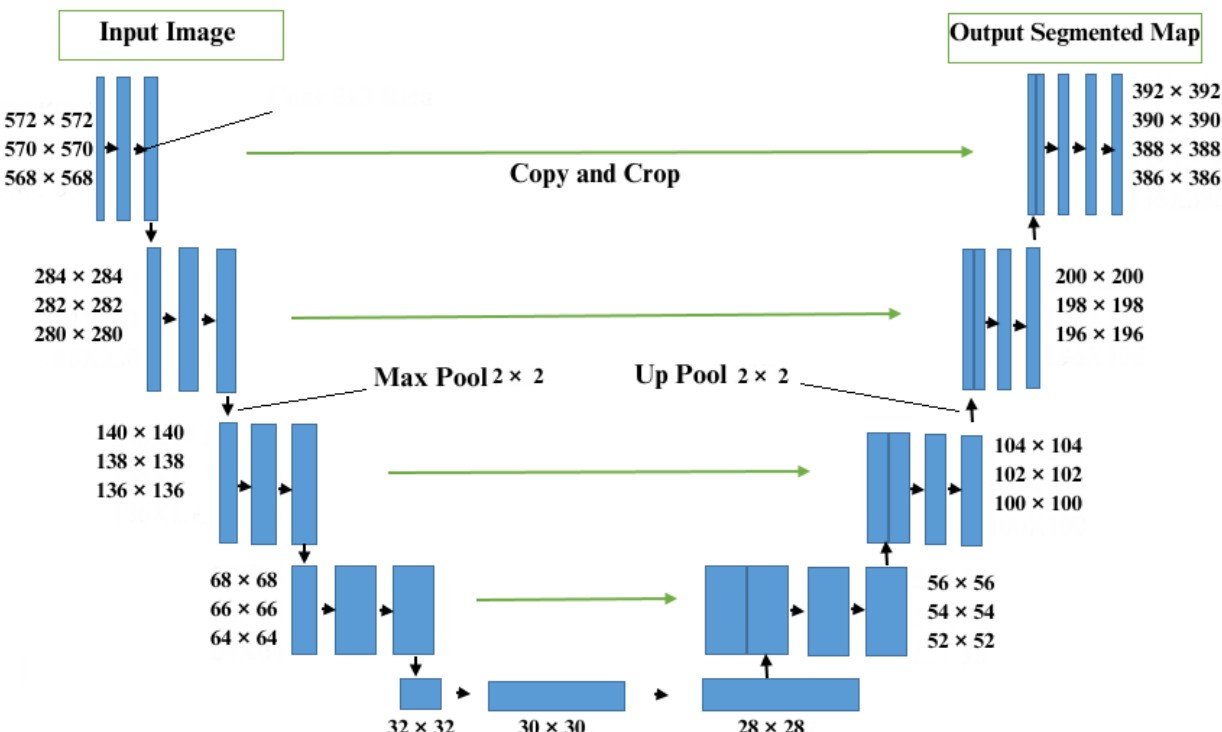

**Figure 4.** Proposed U-Net-based architecture.

The frames from the images are mapped to a segmentation map, as shown in Figure 4, where the driver's body is mapped as one entity, and everything else visible in the image is marked as another entity. Then, the segmentation map is used as a mask to extract the driver's body from each frame. U-Net's name is justified by the shape of its architecture, which resembles that of an autoencoder.

The first half of U-Net captures the context with a compact feature map. The second symmetric half is there for precise localization to retain spatial information, compensating for the downsampling and max-pooling performed in the first stage. The convolution layers in our model (Conv. Nets) embed the information from the frames to connect the last LSTM layers to make a meaningful classification. Sometimes, the output from the Conv. traps becomes vast in the channel dimension, which creates the requirement for relatively more computation in the later stages. This can be solved by using $1 \times 1$ Conv. filters. As shown in Figure 5, the $32 \times 32 \times 512$ output can be combined with a $1 \times 1 \times 512$ filter to reduce the output dimensions to $33 \times 32 \times 1$. The reduced output from the $1 \times 1$ convolution layer is fed into an LSTM layer.

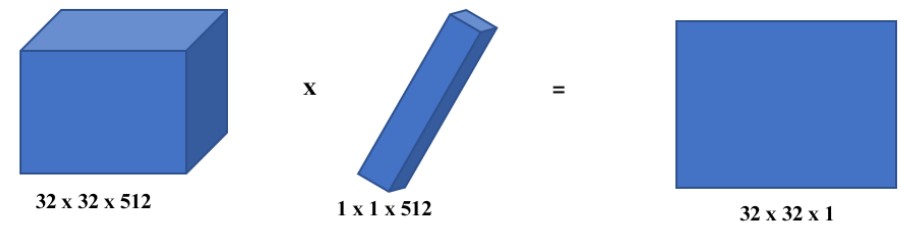

**Figure 5.** Application of $1 \times 1$ filters.

The overall effectiveness of the segmentation results directly contributes to the accuracy and reliability of the entire driver drowsiness detection system. Regular testing and evaluation on diverse datasets and under various conditions are essential to ensure that the segmentation process meets the desired performance standards.

### 4.1. Dataset and Data Preprocessing

We collected the database developed by I. Nasri et al. from Kaggle. The dataset's size was around 5.3 GB, and the total dataset time was approximately 7.3 h. The system obtained frontal pictures of the driver, corresponding to the preceding 55 s captured at 12 frames per second by a camera mounted on the car. The videos featured a frame rate ranging from 12 to 27 frames per second with a resolution of $640 \times 480$. We constructed the experimental environment consisting of more than 3144 photos. We intended to train and test across four different scenarios. For open eyes, we trained using 489 images and tested using 243 images. For closed eyes, we trained using 384 images and tested using 266 images. For open mouths, we trained using 496 images and tested using 320 images. The feature data in this paper vary significantly in scale. In this work, the dataset was normalized to remove the impact of the hierarchy between the features. The normalization procedure aids in increasing computational accuracy, preventing gradient explosion during network training, and hastening the loss function's convergence.

### 4.2. Feature Extraction

Both CNNs and LSTM are well-known deep learning models. A CNN can extract data features in the spatial dimension through layer-by-layer learning of the local features of the data using hidden layers. LSTM can extract features in the temporal dimension and retain contextual historical knowledge over long periods. The activation function, optimization function, and learning rate of the CNN-LSTM model were set to ReLU, RMSprop, and 0.0001, respectively. The samples were first fed into the CNN in the CNN-LSTM model, and then two sampling operations and four one-dimensional convolution operations were performed. The input sample's tensor was (None, 41, 64), where None stands for the input

sample's size. The resulting tensor (None, 10, 128) was fed into the LSTM following the CNN process. The number of output features was adjusted by altering the number of nodes in the first dense layer, which served as the middle layer for extracting features. Finally, the last fully connected dense layer produced the tensor (None, (5–40)). By modifying the epoch value, learning rate, and number of nodes in the dense layer during model training, the ideal model parameters were determined.

*4.3. Detection of Drowsiness State*

Monitoring the PERCLOS can be part of a system for detecting when a driver may be becoming drowsy, which is crucial for preventing accidents on the road. Drowsiness detection is a complex task, and combining multiple indicators often leads to more accurate results. The PERCLOS, when used in conjunction with other measures, contributes to a more comprehensive and reliable drowsiness detection system. The human body reflects its states automatically. EM-CNN uses these kinds of human physiological reactions to evaluate the PERCLOS and POM. The equation of the PERCLOS, using a percentage, is given below in Equation (1) [52].

$$PERCLOS = (\sum_{i}^{N} f_i / N_f) \times 100\% \tag{1}$$

$\sum_{i}^{N} f_i$ represents the frames of a closed eye per unit of time. $N_f$ is the total number of frames per unit, and $f_i$ represents the frame of the closed eye. To calculate the threshold of drowsiness, a collection of 13 video frames was used to test and evaluate the value of the perceptron learning rule with output scaling (PERCLOS). According to Equation (2), a value of 0.25 or greater means that the eye is in a closed state for a continuous period, which indicates drowsiness. The neural network is trained based on the drowsiness thresholds of the PERCLOS and POM [12]. Recurrent Input Output (RIO) refers to a neural network architecture or a specific type of layer that utilizes recurrent connections. The PERCLOS, in the context of neural networks, is not a commonly used acronym. However, it refers to a neural network architecture or algorithm that combines the perceptron learning rule with output scaling. The perceptron learning rule is a fundamental concept in neural networks, and output scaling refers to adjusting the output of neural network layers to match a desired range or format. The POM, in the context of neural networks, is the "Probabilistic Output Model". This refers to a type of neural network or model designed to provide probabilistic predictions or estimates as outputs. For example, probabilistic neural networks or certain types of Bayesian neural networks can produce probabilistic outputs, which are valuable in tasks like uncertainty estimation or probabilistic classification. MT-CNN extracts the facial features along with the feature points, which helps obtain the ROI of the eyes and mouth. Then, EM-CNN evaluates the states of the eyes and mouth. By observing the uninterrupted image frames, the degrees of eye and mouth closure are calculated when a threshold is matched or exceeded. The segmentation algorithm used in the proposed method (U-Net, [13]), is essentially a series of convolution and ReLU blocks with some max-pooling layers between the first and second half of the convolution and ReLU blocks, followed by some transpose convolution layers. The two halves are also connected with multiple skip connections between them. The convolution, ReLU, and max-pooling layers are also used in the primary model, specifically in the CNN part of the CNN-LSTM architecture [53]. The convolution operation is described by Equation (2) [53].

$$\sum \sum I(ip + p, j + r).K_r(p, r) \tag{2}$$

Here, *I* is the input matrix, and K is the 2D kernel with a size of *p* × *r*. In Equation (3), the convolution blocks also use the ReLU activation function to add non-linearity to the output. The operation of the ReLU can be described as f being a function of *x*.

$$f(x) = maximum(0, x) \tag{3}$$

Max-pooling is the most commonly used method among all the pooling layers. It reduces the number of parameters by sampling the maximum activation value from a patch of the image or the matrix. Max-pooling can be described by Equation (4),

$$P_{i,j} = maximum(f(x) : x = A_{i+m,j+n}) \tag{4}$$

where A is the activation output from the ReLU, and P is the output from the max-pooling layer. The U-Net also uses the transposed convolution operation, which is similar to max-pooling but upsamples the encoding instead. In Equation (5), transposed convolution processes an image with a size of *i* × *i* using a kernel with a size of *k* × *k* and outputs an upsampled matrix with dimensions given by the following formula:

$$(i - 1) \times s - (2 \times p) + (k - 1) + 1 \tag{5}$$

where *s* is the stride of the padding. The operation of the transposed convolution involves multiplying the value of the filter with the encoded matrix to obtain another padded and higher-resolution matrix. This stage presents the output from the LSTM layer, which can be used with another system to create an end-to-end helpful system. For instance, a buzzer may be connected at the end, which is triggered when a driver is detected as drowsy, or in the case of a self-driving car, it could safely park on the side of the road and take measures to wake the driver up. Figure 6 presents the architecture of the CNN-LSTM model.

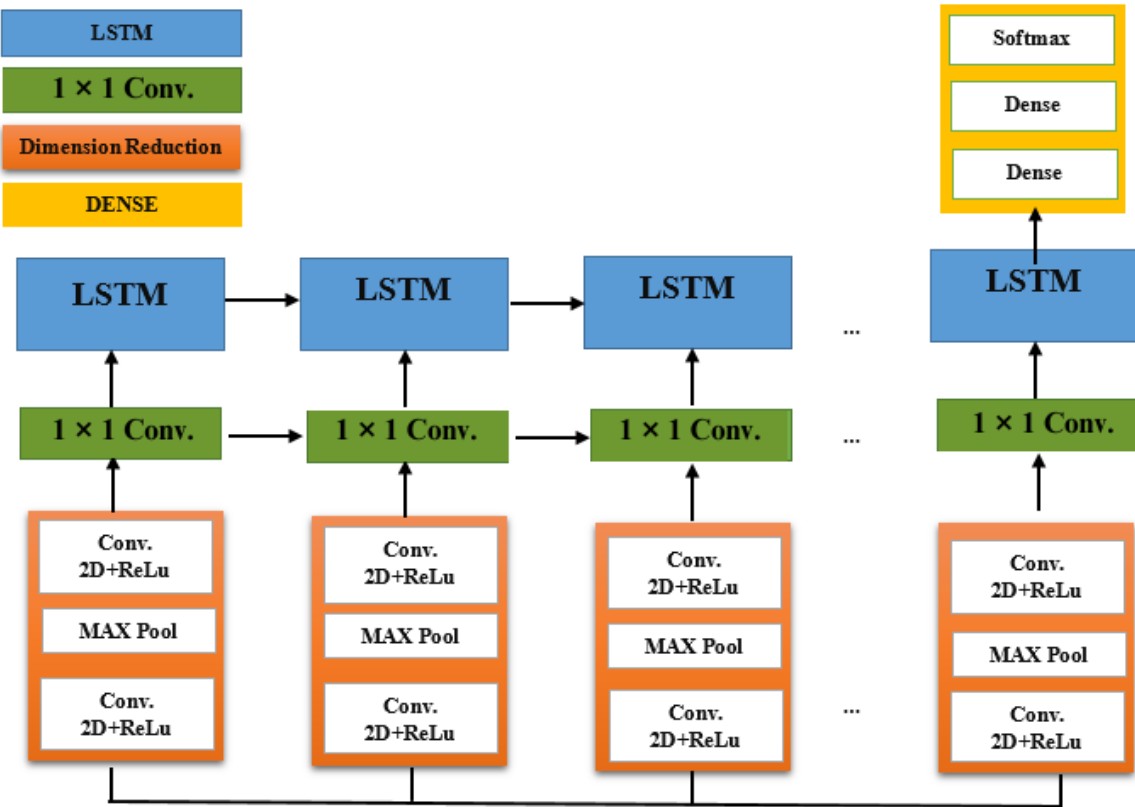

**Figure 6.** CNN-LSTM architecture.

The CNN-LSTM model uses LSTM layers to fuse information from past time steps. For retaining data activation effects for much longer in the recursion, LSTM layers have proven to be more effective compared to GRU layers. The main reason for this difference is that LSTM uses three gates to update the memory cell. One is the update gate, also present in GRU, and the others are the forget and output gates. More formally, the three gates can be described by the following equation [54]:

$$\tau_{update} = \sigma(W_u[a^{t-1}, x^t] + b_u) \tag{6}$$

In the above equation and subsequent equations, $a^{t-1}$ denotes the activation from the previous time step, and $x^t$ denotes the input in the current time step. $W_u$ and $b_u$ represent the parameter matrix and the bias, respectively, and $\tau_{update}$ is the value of the update gate. Then,

$$\tau_{forget} = \sigma(W_f[a^{t-1}, x^t] + b_f) \tag{7}$$

where $W_f$ and $b_f$ are the parameter matrix and the bias, respectively, and $\Gamma_{forget}$ is the value of the update gate.

$$\tau_{output} = \sigma(W_o[a^{t-1}, x^t] + b_o) \tag{8}$$

In (8), $W_o$ and $b_o$ are the parameter matrix and the bias, respectively, and $\Gamma_{output}$ is the value of the update gate.

The memory cell of the LSTM is calculated using the following equation:

$$c^{<t>} = \tau_{update} * c1^{<t>} + \tau_{forget} * c^{<t-1>} \tag{9}$$

where $c^{<t>}$ and $c^{<t-1>}$ are the values of the memory cell. $c1^t$ is the candidate for the memory cell that is supposed to replace the current one. Here, $*$ means vector multiplication. The value for $c1^t$ can be written in terms of the following equation:

$$c1^{<t>} = tanh(W_c[a^{t-1}, x^t] + b_c) \tag{10}$$

In Equation (11), finally, the current activation is calculated by combining the output gate and $c^{<t>}$. Here, $*$ means vector multiplication.

$$a^{<t>} = \tau_{update} * c^{<t>} \tag{11}$$

Detecting drowsiness in drivers is a challenging task that demands a sophisticated approach. One promising solution is the fusion of CNN and LSTM networks, two powerful deep learning architectures. This fusion capitalizes on the strengths of both CNNs, known for their image analysis prowess, and LSTMs, renowned for modeling sequential patterns. The CNN-LSTM architecture holds the potential to revolutionize drowsiness detection systems, making roads safer and saving lives. By analyzing real-time video streams of a driver's face, this hybrid model can not only capture intricate facial features but also track temporal patterns in driver behavior. This innovative approach has the capability to accurately determine when a driver is becoming drowsy, thus enabling timely interventions and preventive measures.

In this system, CNNs are employed as the first line of defense, extracting meaningful features from images of a driver's face. These features are then passed to the LSTM network, which specializes in understanding the sequence of these features over time. By learning from historical patterns, the LSTM can distinguish between normal behavior and signs of drowsiness, such as drooping eyelids, yawning, or erratic facial movements. The strength of this combined architecture lies in its ability to consider not only the current frame but also the context provided by preceding frames. This context-aware capability allows the model to identify subtle changes that might escape a single-frame analysis. As a result, the CNN-LSTM model can adapt to the dynamic nature of drowsiness, which often manifests gradually rather than abruptly. Through rigorous training on diverse datasets encompass-

ing various lighting conditions, driver characteristics, and scenarios, the CNN-LSTM model refines its ability to accurately recognize drowsiness. The model's high accuracy, sensitivity, and specificity make it an indispensable tool for modern driver assistance systems. Its potential applications extend beyond drowsiness detection—it can be integrated into smart vehicles, fleet management systems, and transportation infrastructures, contributing to a safer and more secure transportation ecosystem. The steps of the proposed algorithm are as follows:

Step 1: Preprocess the image (M) datasets.
Step 2: Combine the images with the inputs from the trained models.
Step 3: Retrieve the results of the final convolution layer of the model that was provided.
Step 4: Flatten the n dimensions, decreasing their number to $n-1$.
Step 5: Apply the different layers of CNN-LSTM.

Padding (Conv2d): The formula below is used to determine the padding width, where $pd$ stands for padding, and $fd$ stands for the filter dimension, $fd \in Odd$,

$$pd = \frac{fd-1}{2} \tag{12}$$

Forward propagation: This is separated into two phases. After computing the intermediate value K that is produced through the convolution of the input data from the preceding layer with the M tensor, it then adds bias b and applies a nonlinear activation function on the intermediate values:

$$K^l = M^l \cdot AF^l + b^i, AF^l = g^l(k^l) \tag{13}$$

Max-pooling: The output matrix's proportions can be calculated using (14) while accounting for padding and stride:

$$n_{output} = \frac{n_{output} + 2pd - ft}{s} + 1 \tag{14}$$

The cost function's partial derivative is expressed as

$$\partial AF^l = \frac{\partial l}{\partial AF^l}, \partial K^l = \frac{\partial l}{\partial K^l}, \partial M^l = \frac{\partial l}{\partial M^l}, \partial b^l = \frac{\partial l}{\partial b^l} \tag{15}$$

After applying the chain rule in (15),

$$\partial K^l = \partial AF^l \times g(K^l) \tag{16}$$

The sigmoid activation function, linear transformation, and leaky ReLU are expressed as follows:

$$f(r) = \frac{1}{1+e^{-r}}, K = M^t \cdot R + b, f(r) = (0.01 \times r, r) \tag{17}$$

It returns r if the input is positive and 0.01 times r if the input is negative. As a result, it also produces an output for negative values. This minor modification causes the gradient on the graph's left side to become nonzero.

Applying the softmax function: A neural network typically does not create one final figure. To represent the likelihood of each class, these numbers must be reduced to integers from zero to one.

$$\sigma(m)_j = \frac{e^m j}{\sum_{p-1}^{p} e^m j} \, for \, j = 1..p \tag{18}$$

Applying the CNN-LSTM: LSTM is used after the CNN has been applied, i.e., CNN-LSTM:

$$input_t = \sigma(w_i[h_{t-1}, r_t] + b_i)$$
$$forgetgate_t = \sigma(w_{function}[h_{t-1}, r_t] + b_{forgetgate}) \tag{19}$$
$$output_t = \sigma(w_{output}[h_{t-1}, r_t] + b_{output})$$

where $x_t$ is the input at the current timestamp, and $h_{t-1}$ is the previous LSTM block. $\sigma$ represents the sigmoid function. $forgetgate_t$ is the forget gate. $b$ is the bias for the respective gates. $c_t$ is the cell state at timestamp (t). $\widetilde{c}_t$ represents a candidate for the cell state at timestamp (t):

$$\widetilde{c}_t = tanh \, tanh(w_c[h_{t-1}, r_t] + b_c),$$
$$c_t = f_t * c_{t-1} + i_t * \widetilde{c}_t, h_t = o_t * tanh \, tanh(c_t) \tag{20}$$

In the context of driver drowsiness, this hybrid CNN-LSTM architecture represents a formidable tool. By analyzing real-time video feeds of a driver's face, the CNN component is capable of discerning crucial facial features, such as eye movement, blink rate, and facial expressions. These features, extracted through the convolutional layers of the CNN, serve as a rich foundation of visual cues. Figure 7 presents the architecture of the proposed model. However, what sets this architecture apart is its LSTM component. This network structure possesses the ability to understand sequences and patterns within the extracted features. In the realm of drowsiness detection, this implies that the system can not only assess the current state of the driver's face but also track how that state evolves over time. Subtle cues that might signify drowsiness, such as prolonged eye closure or micro-expressions, can be identified by the LSTM as part of a sequence, enabling more accurate and reliable detection. The predictive power of the CNN-LSTM architecture extends beyond mere real-time analysis. By leveraging the LSTM's memory capabilities, the model can recognize trends and tendencies that indicate an increasing likelihood of drowsiness. This forward-looking approach allows for timely intervention, such as alerts to the driver or automated adjustments to vehicle settings, preventing potential accidents before they occur.

A CNN-LSTM network is a popular architecture used in deep learning for tasks that involve both spatial and temporal data, such as video analysis or sequential image data. The input data for a CNN-LSTM network typically consist of a sequence of images or tensors, where each element in the sequence represents a frame in a video or a timestamp in a time series. The input data are first passed through a CNN to extract spatial features. The CNN layers consist of convolutional layers followed by pooling layers, which help capture important spatial patterns in the data. These layers are responsible for feature extraction from individual frames. After the CNN layers, the features can either be flattened into a 1D vector or global average pooling can be used to reduce the spatial dimensions. This step depends on the specific problem and the architecture used. The output of the CNN is then fed into an LSTM network, which is responsible for capturing temporal dependencies and patterns across the sequence of frames. The LSTM network consists of multiple LSTM layers. Each LSTM cell maintains an internal state that can capture information from previous time steps. This internal state helps model long-term dependencies in the data. The LSTM layers process the sequence of features generated by the CNN, one time step at a time, and update their internal states accordingly. The final LSTM layer is often connected to one or more fully connected (dense) layers, which can be used for making predictions or classifications. The output layer's architecture depends on the specific task. For example, video classification might consist of a softmax layer for class probabilities. In our approach, the MSE (mean-squared error) function was utilized as the loss function. To update the settings of each network layer, the common Adam optimization technique was

employed as the optimizer. The dropout layer we used contributed to the model's enhanced generalizability, decreased the training time, and prevented overfitting. In our research, the constructed model's prediction performance was compared with that of the EM-CNN, VGG-16. GoogLeNet, AlexNet, and ResNet50 models in order to confirm the model's efficacy. These methods were chosen for comparison because of their specific characteristics. EM-CNN is a semi-supervised learning algorithm that uses only weakly annotated data and performs very efficiently in face detection. VGG-16 is a 16-layer deep neural network, a relatively extensive network with a total of 138 million parameters, that can achieve a test accuracy of 92.7% on ImageNet, a dataset containing more than 14 million training images across 1000 object classes. GoogLeNet is a type of CNN based on the Inception architecture. It utilizes Inception modules, which allow the network to choose between multiple convolutional filter sizes in each block. AlexNet uses an 8-layer CNN, showing, for the first time, that the features obtained through learning can transcend manually designed features, thereby breaking the previous paradigm in computer vision. ResNet-50 is a 50-layer CNN (48 convolutional layers, 1 max-pooling layer, and 1 average-pooling layer) that forms networks by stacking residual blocks.

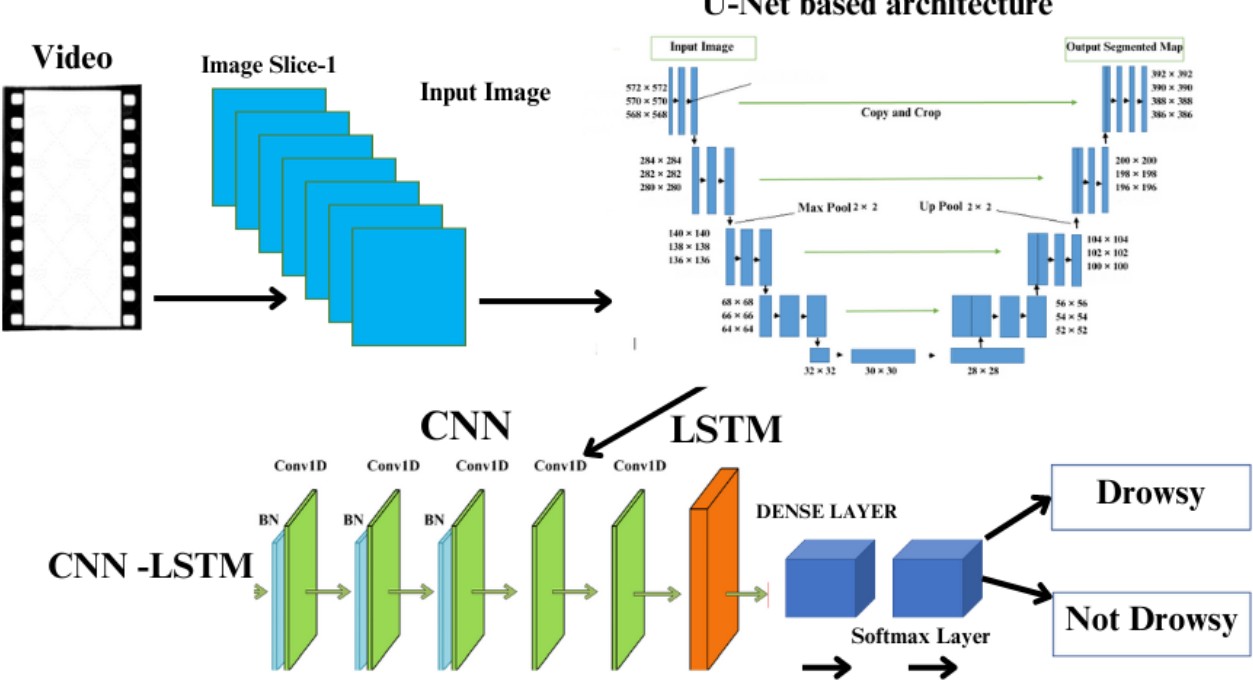

**Figure 7.** Proposed model architecture.

## 5. Experimental Results

We aimed to identify drowsiness and awaken users to prevent accidents by producing an alarm sound and app notification. The proposed approach yielded results that were greater than 98% in terms of accuracy. For this project, we needed real-world images of drivers while driving. This real-world environment helps in building the architecture, resulting in an accurate model. Figure 8 represents the training and validation accuracy of the training dataset.

Last but not least, the mouth in the closed state was represented by 640 images for training and 475 for testing. Some sample images depicting routine and drowsy states in both daylight and nighttime are shown in Figures 9 and 10.

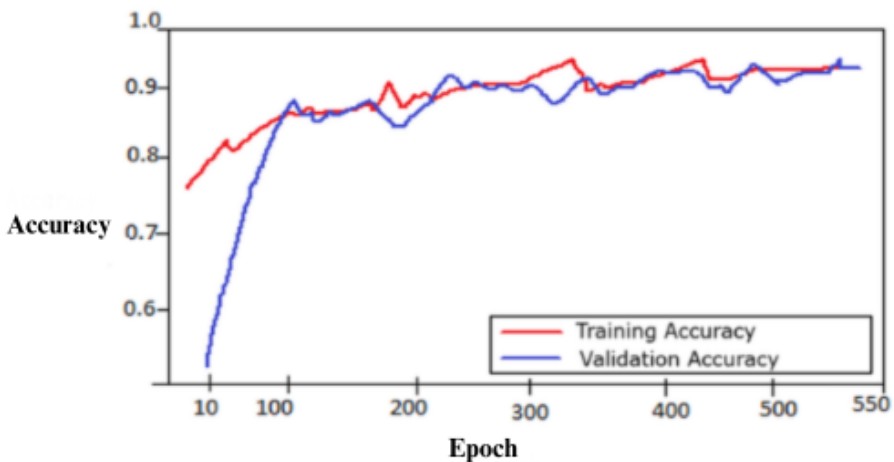

**Figure 8.** Training and validation accuracy.

**Figure 9.** Sample normal and drowsy states (daylight).

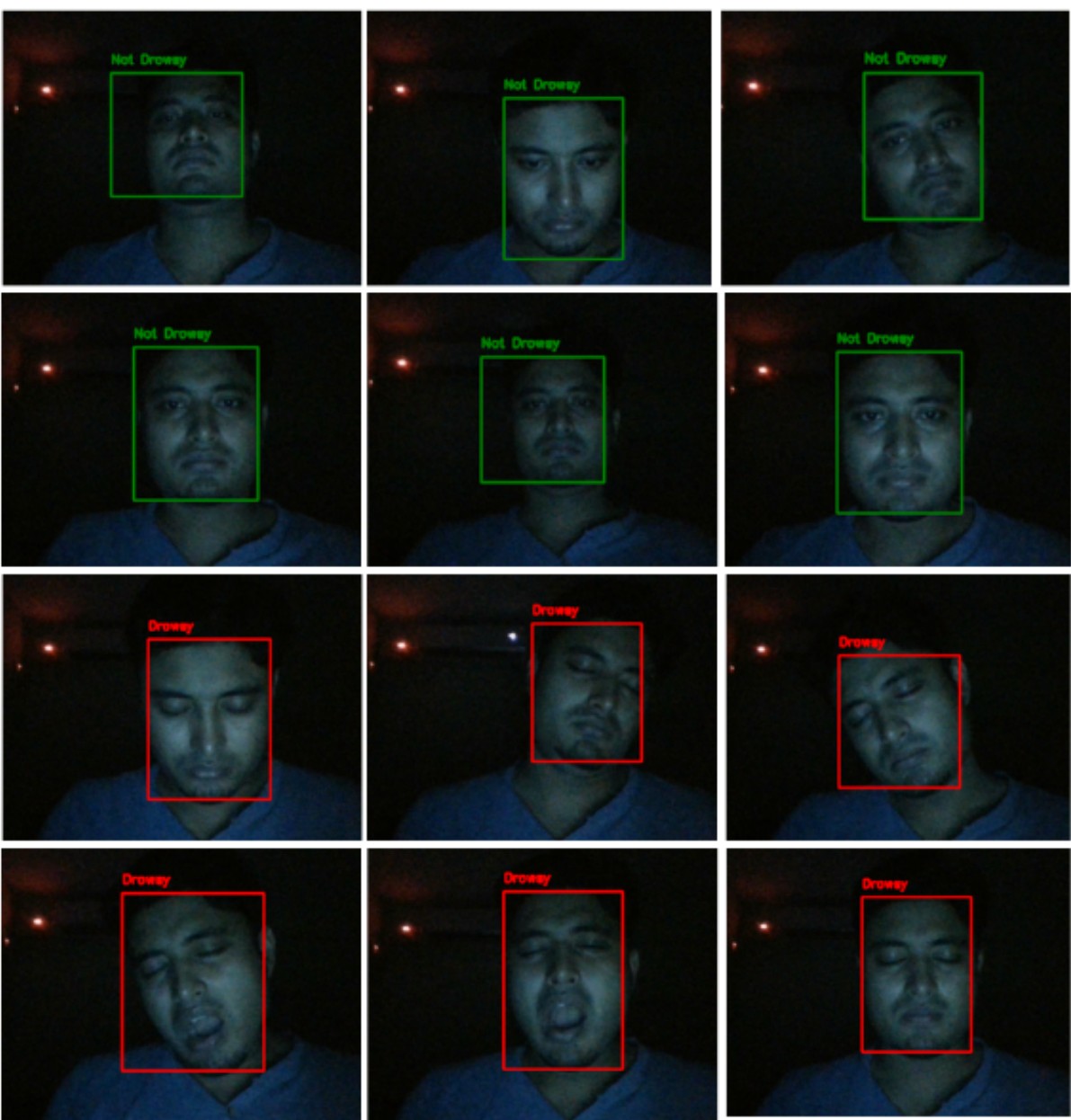

**Figure 10.** Sample normal and drowsy states (nighttime).

*5.1. Implementation Process and Results Discussion*

Emotion-CNN (EM-CNN) represents a cutting-edge approach to emotion recognition, leveraging the power of convolutional neural networks (CNNs) to decode and understand the nuanced expressions of human emotions within visual data. In a world where artificial intelligence continues to bridge the gap between human experiences and computational capabilities, EM-CNN has emerged as a specialized architecture designed to decipher the complex language of emotions embedded in images and videos. The VGG-16 architecture has emerged as a pivotal milestone in the evolution of convolutional neural networks (CNNs) for image classification and computer vision. Conceived by the Visual Geometry Group at the University of Oxford, VGG-16, which stands for Visual Geometry Group with 16 layers, represents a breakthrough in the pursuit of deep learning excellence. VGG-16 contains 16 weight layers, including 13 convolutional layers and 3 fully connected layers. It is known for its simplicity, having small $3 \times 3$ convolutional filters and a deep architecture, which aids in feature learning.

In the dynamic landscape of deep learning, GoogLeNet, or Inception v1, represents a groundbreaking convolutional neural network architecture developed by researchers at Google. Introduced in 2014, it was designed to address challenges associated with computational efficiency, parameter reduction, and the ability to capture diverse features across multiple scales. GoogLeNet is deeper than VGG-16 but uses a more efficient architecture with 1 × 1 convolutions for dimension reduction and parallel filter operations. This reduces the number of parameters and computational cost. AlexNet is a deep convolutional neural network (CNN) featuring eight layers. The architecture's depth was a departure from previous shallower networks, allowing it to learn intricate hierarchical features from raw image data. AlexNet consists of five convolutional layers followed by max-pooling layers and three fully connected layers. It helped establish the effectiveness of deep learning in computer vision tasks. ResNet50, a variant of the ResNet (Residual Network) architecture, stands as a monumental advancement in deep neural networks, particularly in addressing the challenges associated with training very deep networks. The defining feature of ResNet50 lies in its introduction of residual learning. In traditional deep networks, the optimization process can be hindered by the vanishing gradient problem, making it challenging for information to flow through the network. Residual learning addresses this by introducing shortcut connections, or skip connections, allowing the network to learn residual functions. ResNet50's skip connections mitigate the vanishing gradient problem, allowing the training of very deep networks. This architecture is widely used in image classification and other computer vision tasks.

CNN-LSTM, a hybrid neural network architecture, represents a sophisticated integration of convolutional neural networks (CNNs) and Long Short-Term Memory (LSTM) networks. This fusion is designed to harness the strengths of both convolutional and recurrent architectures, making it particularly well suited for tasks that involve both spatial and temporal dependencies. In many real-world applications, data are not only spatially rich but also exhibit temporal dependencies. For instance, in video analysis, understanding the content requires capturing both the spatial features (such as objects and patterns within frames) and temporal dynamics. CNN-LSTM has emerged as a solution to address this dual challenge. The CNN part is used for spatial feature extraction, capturing patterns in different spatial regions, whereas the LSTM part handles temporal dependencies by processing sequences of features. One thing that is easy to identify is that when a person is driving and maintaining an average amplitude, there is less variation in the mouth state. But on the other hand, it is harder to define an eye-blinking state, as machine vision calculates the closure state. To simplify this process, a detailed flowchart is provided to help clarify the idea. By applying Erosion and Expansion binaries, the ROI is smoothed, which helps define the state of the eye. Black pixels allow for representing the binocular image area. The count of black pixels is also essential. The threshold value was set to 0.15, which defines the eye-closure state. A value exceeding 0.15 is considered an eye in an open state. A batch size of 32 was used for our training. Selecting the number of epochs and the batch size for drowsiness detection using a neural network involves considering various factors related to the dataset, model architecture, and computational resources. Drowsiness detection often involves intricate patterns and temporal dependencies. The number of epochs should be sufficient to allow the model to learn and capture these complex patterns in the data. We experimented with different epoch values and observed the model's behavior on both the training and validation sets. Drowsiness detection often involves analyzing sequences of data over time, such as video frames or time-series data from sensors. Smaller batch sizes might allow the model to capture temporal dependencies more effectively. If the drowsiness detection system is intended for real-time applications, the batch size must balance model accuracy and inference speed. Smaller batch sizes can lead to faster predictions, which is crucial in real-time scenarios. The selection of the number of epochs and the batch size for drowsiness detection involves a balance between capturing temporal dependencies, computational efficiency, and preventing overfitting. Experimentation and close monitoring of model performance are essential for making informed decisions. It

takes special consideration to implement a neural network on hardware. Both training and testing can be carried out with a large amount of RAM and a powerful processor. During our training, we trained our model with 100 epochs. The number of epochs and the batch size were chosen empirically, as shown in Figure 8, as the best trade-off between the accuracy level and the computational complexity required by the investigated algorithm. The system configuration comprised a computer with six-core processors, 16 GB of RAM, and an Nvidia GTX 1650Ti GPU running on a 64-bit Windows 10 system. This processing power was adequate for the operation of our application. The suggested approach used Google's "Colab Pro Plus version" as the execution platform. Conversely, MT-CNN, EM-CNN, and CNN-LSTM were implemented using Python 3.10 and Keras 2.4.0, with Tensorflow 2.70 as the environment.

We can see that the training accuracy values achieved by EM-CNN, VGG-16, GoogLeNet, AlexNet, ResNet50, and our proposed model were 86.54%, 92.46%, 66.19%, 46.12%, 56.09%, and 98.70% and the testing accuracy values achieved were 89.54%, 92.4%, 66.19%, 48.50%, 56.09%, and 98.80%. Table 3 presents the training accuracy and testing accuracy values of the different networks.

**Table 3.** Experimental training and testing results.

| Network | Training Accuracy (%) | Testing Accuracy (%) |
|---|---|---|
| EM-CNN | 86.54 | 86.54 |
| VGG-16 | 92.46 | 92.46 |
| GoogLeNet | 66.19 | 66.19 |
| AlexNet | 46.12 | 48.5 |
| ResNet50 | 56.09 | 56.09 |
| **Proposed Model (CNN-LSTM)** | **98.7** | **98.8** |

Table 4 presents the precision, recall, F1 score, and accuracy values of GoogLeNet, ResNet50, AlexNet, VGG-16, EM-CNN, and CNN-LSTM. CNN-LSTM yielded better results compared to the other learning algorithms. TensorFlow was then used to translate the 10.5 h of video data in the dataset into frames. The research employed various metrics to evaluate how well deep learning models could detect driver sleepiness. These metrics encompassed accuracy, loss, precision, recall, and F1 score. To elucidate a model's performance, a confusion matrix is frequently used, as depicted in Figure 11. This matrix serves as a table to gauge the accuracy of a deep learning model across different dataset types using a test dataset. In the pursuit of training robust and accurate models for drowsiness detection, the optimization of loss functions serves as a crucial compass, guiding the neural network toward learning the intricate patterns indicative of drowsiness. The selection of an appropriate loss function is akin to fine-tuning the model's compass, aligning it with the landscape of the drowsiness detection task. Here, we explore the essence of this journey, understanding the key considerations and pathways in loss function optimization. In the grand expedition of drowsiness detection, the optimization of loss functions becomes a precision compass, guiding the neural network through the diverse and challenging terrains of imbalanced data, temporal dependencies, and nuanced pattern recognition. With each epoch, the model refines its navigation skills, inching closer to the destination of heightened accuracy and vigilance in drowsiness detection. The loss function defines the objective that the model aims to minimize during training. The experiment's loss function is categorical cross-entropy, which is expressed as:

$$loss = -\sum_{i=1}^{N} x_{a,b} In p_a, b \tag{21}$$

where $N$ is the number of classes; $x$ is a binary indication indicating whether or not c is the accurate prediction for observation $a$; and $p$ is the anticipated probability that the

observation belongs to class *b*. Each deep learning model completes roughly 35 epochs throughout the fitting process, with a batch size of 32.

**Table 4.** Results of deep learning models in terms of precision, recall, F1 score, and accuracy

| Network | Precision | Recall | F1 Score | Accuracy (%) |
|---|---|---|---|---|
| EM-CNN [55] | 0.721 | 0.685 | 0.602 | 96.62 |
| VGG-16 [56] | 0.113 | 0.218 | 0.117 | 82.98 |
| GoogLeNet [19] | 0.708 | 0.611 | 0.594 | 94.01 |
| AlexNet [57] | 0.461 | 0.485 | 0.312 | 91.48 |
| ResNet50 [58] | 0.677 | 0.536 | 0.514 | 93.85 |
| **Proposed Model (CNN-LSTM)** | **0.819** | **0.652** | **0.732** | **98.46** |

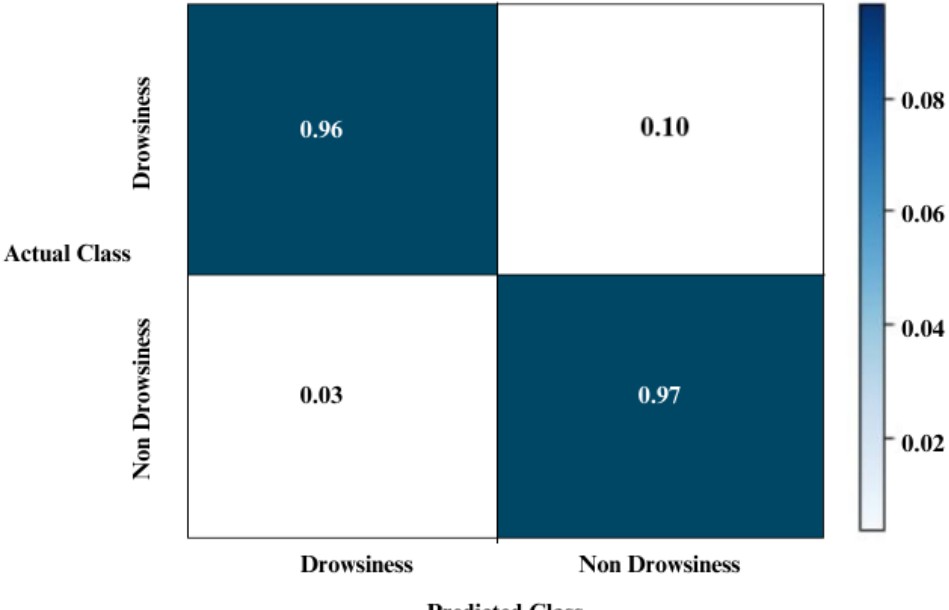

**Figure 11.** Confusion matrix.

The confusion matrix for driver drowsiness detection using a CNN-LSTM model is a table that shows the performance of the model in terms of true positives (TP), true negatives (TN), false positives (FP), and false negatives (FN). It helps evaluate the accuracy of the model in classifying instances of driver drowsiness.

A true positive (TP) correctly predicts a drowsy driver. A true negative (TN) correctly predicts a non-drowsy driver. A false positive (FP) incorrectly predicts a driver as drowsy when they are not. A false negative (FN) incorrectly predicts a non-drowsy driver as drowsy. In Equation (22), precision measures the accuracy of positive predictions. It is the ratio of true positives to the total number of instances predicted as positive.

$$\text{Precision} = \frac{\text{TP}}{\text{TP} + \text{FP}} \tag{22}$$

In Equation (23), recall represents the ability of the model to capture all relevant instances. It is the ratio of true positives to the total number of actual positive instances.

$$\text{Recall} = \frac{\text{TP}}{\text{TP} + \text{FN}} \tag{23}$$

In Equation (24), the F1 score represents the harmonic mean of precision and recall. It provides a balanced measure that considers both false positives and false negatives. It is particularly useful when there is an imbalance between classes.

$$F1\ Score = \frac{2 \times Precision \times Recall}{Precision + Recall} \tag{24}$$

In Equation (25), accuracy represents the overall correctness of a model. It is the ratio of correctly predicted instances (both true positives and true negatives) to the total number of instances.

$$Accuracy = \frac{TP + TN}{TP + TN + FP + FN} \tag{25}$$

Accuracy is the percentage of samples correctly categorized by the classifier among all samples within a given test dataset, or the test dataset's accuracy when the loss function is 0–1. The loss function is used to gauge how well a model predicts, and the lower it is, the better. The class in question is typically regarded as the positive class, whereas other classes are regarded as the negative class. Sensitivity measures the ability of a classification model to correctly identify true positive instances among all actual positive instances. Specificity gauges the ability of a classification model to correctly identify true negative instances among all actual negative instances.

$$Sensitivity = \frac{TP}{TP + FN} = Recall \tag{26}$$

$$Specificity\ (\%) = \frac{TN}{TN + FP} \tag{27}$$

When we compare a model with others, we can appreciate its efficiency. We compared our CNN-LSTM models with other methods like GoogLeNet, ResNet50, AlexNet, VGG-16, and EM-CNN. After comparing the entire procedure, the EM-CNN model proved its efficiency by outperforming the other models with 97.46% accuracy, 97.67% sensitivity, and 78.21% specificity. We can see the specificity (%) of all techniques in Figure 12.

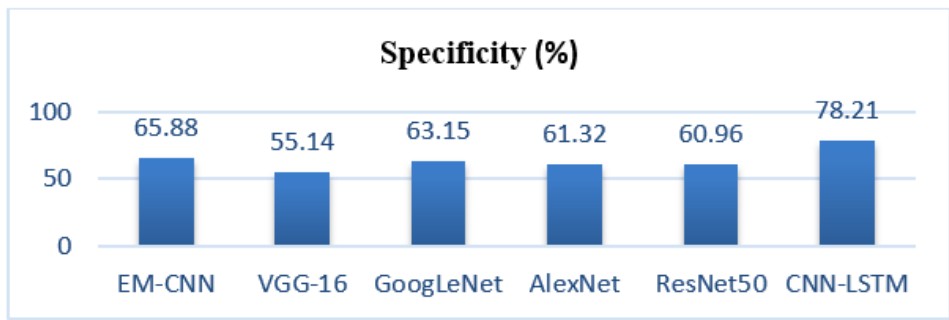

**Figure 12.** Specificity (%) of the different deep learning models

Figure 13 depicts the loss values of a single scenario after numerous training steps (epochs) with learning rates of $1 \times 10^{-4}$ and $1 \times 10^{-5}$. The loss value is shown on the y-axis, and the epochs are represented on the x-axis. The accuracy steadily improved as the time lengthened and the loss value dropped. We finished training and continued to the testing phase when the epochs approached 35. The loss values of the different deep learning models are presented in Table 4.

After thorough testing and comparison, we can conclude that CNN-LSTM is more accurate and sensitive to the state of the mouth compared to the state of the eyes. Also, the mouth displays more precise indications for drowsiness, which is a good sign. In [59], the AUC for CNN-LSTM classifications was as follows. Using a temporal correction system enables the detection of eye-blink frequency from video frames. In this system, when an eye is open it is represented by 1, and when closed, it is represented by 0, creating a sequence of 1s and 0s for blinking frequency. Now, it is time to apply the threshold trigger, as we cannot rely solely on the current results.

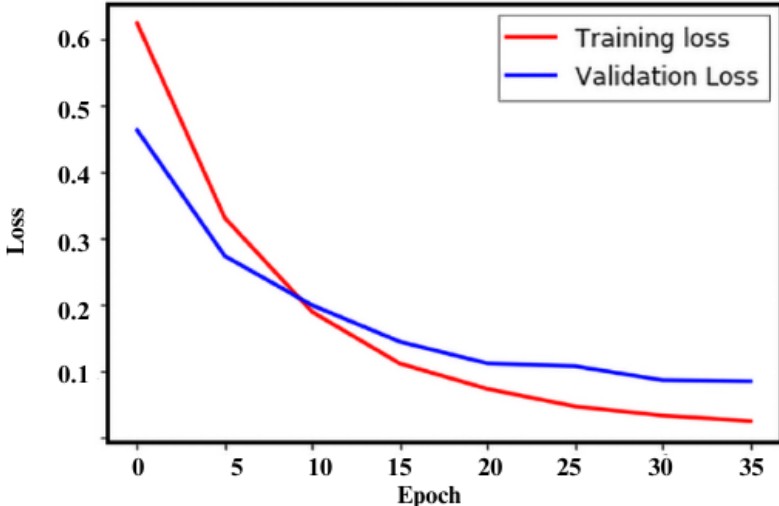

**Figure 13.** Loss value.

For this project, the threshold values of the PERCLOS as POM must be extracted. For frame-by-frame recognition, we extracted images from 15 video frame sequences. The accuracy in terms of the AUC achieved by the CNN-LSTM model in classifying different states, such as eye open, eye closed, mouth closed, and mouth open, is presented in Table 5. Table 5 presents the accuracy and AUC of the CNN-LSTM model on different datasets. AlexNet models were utilized for comparison. The model size of the proposed CNN-LSTM was 21.77 MB, making it 25% smaller compared to the EM-CNN and other algorithms. Compared to previous approaches, the proposed CNN-LSTM model is significantly smaller, simpler, and requires less storage.

**Table 5.** Accuracy (%) and AUC (%) of the proposed CNN-LSTM model in classifying various states on different datasets.

| Dataset (Images) | Accuracy | Eye Closed | Eye Open | Mouth Closed | Mouth Open |
|---|---|---|---|---|---|
| 5 | 98.32 | 99.22 | 99.11 | 99.65 | 99.34 |
| 15 | 97.34 | 95.72 | 94.28 | 96.95 | 99.56 |
| 35 | 98.43 | 97.32 | 99.21 | 99.35 | 99.34 |
| 65 | 98.12 | 97.37 | 94.29 | 96.95 | 98.94 |
| 95 | 98.33 | 98.32 | 99.29 | 99.45 | 99.24 |
| 115 | 98.44 | 94.89 | 96.91 | 97.69 | 92.38 |
| 135 | 98.34 | 98.32 | 99.21 | 99.45 | 99.34 |
| 165 | 98.22 | 98.61 | 96.67 | 98.91 | 98.76 |
| 195 | 98.22 | 97.39 | 94.73 | 96.74 | 96.14 |
| 235 | 97.11 | 98.32 | 99.24 | 99.45 | 99.24 |
| 265 | 97.18 | 94.89 | 96.56 | 97.69 | 92.38 |
| 285 | 97.22 | 98.32 | 99.67 | 99.45 | 99.34 |
| 345 | 97.26 | 98.62 | 99.21 | 99.68 | 99.14 |
| 365 | 98.43 | 98.32 | 99.21 | 99.45 | 99.34 |
| 385 | 98.12 | 98.61 | 96.67 | 98.91 | 98.76 |
| 400 | 98.33 | 97.39 | 94.73 | 96.74 | 96.14 |
| 415 | 97.35 | 97.31 | 94.35 | 96.47 | 98.34 |
| 425 | 98.11 | 98.32 | 99.21 | 99.45 | 99.34 |
| 445 | 97.35 | 98.32 | 99.20 | 99.45 | 99.34 |
| 455 | 98.43 | 98.34 | 99.27 | 99.45 | 99.35 |
| 465 | 98.12 | 98.34 | 99.21 | 99.56 | 99.39 |

Table 6 shows the overall speed, drowsiness detection time, and compression of different deep learning models for driver drowsiness. This study focused on estimating

driver tiredness using videos recorded while the driver was on the road. We tested the proposed prediction models on an established dataset.

**Table 6.** The overall speed, drowsiness detection time, and compression of different deep learning models.

| Parameters | Workstation Environment | EM-CNN | VGG-16 | GoogLe-Net | Alex-Net | ResNet-50 | Proposed CNN-LSTM |
|---|---|---|---|---|---|---|---|
| Compression (MB) | Lenovo workstation | 33.6 | 2134 | 1265 | 1998 | 984 | 21.77 |
| Drowsiness detection time (seconds) | Lenovo workstation | 66.7 | 88.34 | 96.65 | 56.87 | 89.90 | 26.88 |
| Overall speed (fps) | Lenovo workstation | 12.4 | 12.1 | 14.67 | 28 | 15.67 | 11.6 |

Table 6 shows that the model's accuracy under these circumstances was highly accurate. The videos with "No Glasses", which depict ideal road conditions, were the most accurate. Sunglasses blocked the driver's vision and lowered the quality of the characteristics the model could detect; hence, the classification accuracy was the lowest. There are other features, such as the shape of the lips, the axis of the head, and so on, in addition to the eyes that can be considered. Analytical analysis of the main characteristics that the CNN and LSTM models automatically transform into dynamic actions allows for a conclusion. All of the evaluation parameters were significantly improved using the proposed CNN-LSTM model. Therefore, the overall performance enhancements resulting from fusing the CNN model with LSTM encourage its implementation in real-time applications. The proposed method achieved an accuracy of 98.46%. The optimization objectives for the loss functions in "Driver Drowsiness using CNN-LSTM and U-Net" involve setting up appropriate loss functions for binary classification (CNN-LSTM) and pixel-wise semantic segmentation (U-Net) and possibly combining these losses in a balanced manner when integrating the two models. The effective choice and tuning of these loss functions are critical for training a model that can accurately detect driver drowsiness based on visual cues. When working on the task of driver drowsiness detection using a combination of CNN-LSTM and U-Net, setting the appropriate loss functions is a crucial step in optimizing the neural network models. Loss functions quantify the error between the predicted outputs and ground-truth labels, and their choice impacts the training and performance of the models. This paper's contribution extends to the novel data acquisition methodology it employs. By capturing a range of facial movements and expressions, including eye closure duration, blinking patterns, and head orientation, the system acquires real-time data that are crucial for accurate drowsiness detection.

### 5.2. Limitations and Constraints

Binarization does not function well for those with dark skin. Binarization methods often rely on contrast between foreground and background. Darker skin tones may have lower contrast under certain lighting conditions, making it challenging for standard binarization techniques to accurately separate features. Traditional binarization methods may not be sensitive to color variations in different skin tones. Grayscale-based methods may not capture the diversity of skin colors effectively. If the binarization model is trained on a dataset that lacks diversity in skin tones, it may not generalize well to individuals with dark skin. Addressing the issue of binarization not functioning well for individuals with dark skin requires a holistic approach involving technical improvements, data diversity, ethical considerations, and community engagement. It is essential to strive for fairness, inclusivity, and accuracy in image processing algorithms to avoid perpetuating biases and limitations. There cannot be any reflective materials behind the driver, which is another restriction. The

system becomes more reliable as the background becomes more homogeneous. A black sheet was placed behind the test participant to solve this issue for testing purposes. Rapid head movement was not permitted throughout testing. This can be compared to emulating a weary driver, which was acceptable in this context. Head motions were rarely missed by the system. The videos that included "No Glasses", which depicted perfect driving circumstances, were the most accurate. The classification accuracy was the lowest with sunglasses on, which blocked the driver's eyesight and reduced the quality of the traits the model could identify. In addition to the eyes, there are additional characteristics that could be considered, such as the form of the lips and the axis of the head. So, by analyzing the significant properties that the CNN and LSTM models automatically convert into dynamic actions, conclusions can be drawn. This is obvious since the system's algorithm is fundamentally dependent on binarization.

Anomaly detection techniques could be implemented to identify instances where the model might struggle due to unusual conditions, such as extremely low light. Different hyperparameter settings could be experimented with to optimize performance, including the learning rate, batch size, and model architecture. Furthermore, we must ensure that our model can handle challenging situations like glare, reflections, or unusual headlight shapes in real scenes.

CNN-LSTM with U-Net is designed to analyze both the spatial and temporal aspects of sequential data, as well as perform image segmentation tasks, whereas sequential order aware coding based robust subspace clustering focuses on clustering and coding techniques for human action recognition in untrimmed videos; GAN siamese network for cross domain vehicle re identification focuses on domain adaptation and similarity learning for vehicle recognition in intelligent transport systems; and Spatio temporal feature encoding for traffic accident detection in VANET environment focuses on encoding and analyzing spatio-temporal patterns for traffic accident detection in vehicular communication networks. The proposed work relates to neural network architectures and their applications in computer vision and deep learning, whereas ann efficient and secure identity based signature system for underwater green transport system pertains to cryptography and secure communication within the context of underwater transportation systems. These concepts are distinct in terms of their nature, purpose, and application domains.

## 6. Conclusions

Current advancements in road safety measures have been considerably propelled by the integration of Internet of Things (IoT) technology with facial movement analysis for the purpose of autonomous driver sleepiness detection. Recently, the discipline of deep learning has resolved many key issues. This study discusses a methodology for the detection of driver drowsiness through the utilization of real-time monitoring. In order to detect driver tiredness, the present study has devised a deep learning model utilizing a CNN–Long Short-Term Memory architecture. Various methods, including EM-CNN, VGG-16, GoogLeNet, AlexNet, and ResNet50, were utilized for comparison, and it is evident that the CNN-LSTM approach demonstrates superior performance compared to the other deep learning techniques. More testing will be required to produce accurate results on its performance with the portability feature, enabling further use with hardware supplies. In the near future, it will be advantageous to mitigate the potential hazards associated with accidents resulting from driver drowsiness. Regarding future research, our model could potentially benefit from the incorporation of an attention module. Improving the model's performance in drowsiness detection would involve a combination of optimizing the model architecture, fine-tuning hyperparameters, addressing data-related challenges, and incorporating advanced techniques. Attention modules play an important role in the human vision perceptron; they can allocate available resources to selectively focus on processing the salient part instead of the whole scene, capturing long-range feature interactions, and boosting the representation capability for the CNN. This addition would enhance

the model's performance by allowing it to consider more nuanced features throughout the categorization process.

**Author Contributions:** Conceptualization, S.D., S.P. and B.P.; Methodology, S.D., S.P., B.P. and R.H.J.; Software, S.D., S.P. and B.P.; Validation, S.D., S.P. and B.P.; Formal analysis, R.H.J.; Investigation, S.D.; Writing—original draft, S.D., S.P., B.P. and R.H.J.; Writing—review & editing, R.H.J. and F.B.; Supervision, F.B. All authors have read and agreed to the published version of the manuscript.

**Funding:** This research received no external funding.

**Data Availability Statement:** The data presented in this study are available on request from the corresponding author. The data are not publicly available due to privacy.

**Conflicts of Interest:** The authors declare no conflict of interest.

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
