# Peer review of "IoT-Assisted Automatic Driver Drowsiness Detection through Facial Movement Analysis Using Deep Learning and a U-Net-Based Architecture"

_information, doi:10.3390/info15010030_

Round 1
Reviewer 1 Report (Previous Reviewer 2)
Comments and Suggestions for Authors
-
There are some points are still needed to be revised as follows:
Clarification on Models and Algorithms: The paper discusses various models such as EM-CNN, VGG-16, GoogleNet, AlexNet, ResNet50, and CNN-LSTM. However, there seems to be a lack of detailed explanation or distinction between these models. It would be beneficial to include brief descriptions of each model and highlight their unique features or aspects that make them suitable for the study. -
Technical Details and Parameters: The paper mentions the use of deep learning models completing about 35 epochs with a batch size of 32, but it doesn't elaborate on the choice of these parameters. Adding justification for the selection of these specific parameters (epochs and batch size) would enhance the understanding of the model's configuration.
-
Discussion on Loss Function Optimization: The paper discusses the optimization of loss functions for binary classification (CNN-LSTM) and pixel-wise semantic segmentation (U-Net). More detailed discussion on why these particular loss functions were chosen and how they impact the model's performance would provide a deeper insight into the model's design choices.
-
Exploration of Limitations and Constraints: The document briefly touches on limitations such as binarization not functioning for those with dark skin and the need for a homogenous background. Expanding on these limitations, possibly in a dedicated section, would provide a more comprehensive understanding of the constraints and practical challenges faced by the system.
-
Enhancements to Future Work Section: The conclusion mentions the potential incorporation of an attention module in future research to improve model performance. Elaborating on how this module could specifically enhance the model and what kind of attention mechanisms are being considered would add value to the future work section.
-
Reference Formatting: Some references seem to be inconsistent in formatting (e.g., missing publication years or inconsistent citation styles). To broaden the scope of this paper, the authors may refer to some work such as I-Health: SDN-based fog architecture for IIoT applications in healthcare and Blockchain and PUF-based lightweight authentication protocol for wireless medical sensor networks. Standardizing the references according to the journal's guidelines would improve the paper's professionalism.
-
Clarification on Data Acquisition Methodology: The paper states a novel data acquisition methodology involving a range of facial movements and expressions. More details on this methodology, including how the data was captured and processed, would provide clarity and strengthen the methodological aspect of the paper.
-
Graphical Representations: While the paper includes figures and tables, some of the graphical representations could be improved for better clarity. For instance, ensuring that all figures and tables are clearly labeled and referenced in the text would enhance the paper's readability.
none
Author Response
Please see the attachment.

Reviewer 2 Report (Previous Reviewer 3)
Comments and Suggestions for Authors
1、Some parts of this article lack references, such as: “If you look into recent times...”.
2、In Figure 8, I did not find any drawings related to 'Sample Normal and Drowsiness Images, Different states of the training dataset'.
3、In the experimental section, I found that you have too much space to describe U-Net and other networks. Please provide more information about the experimental situation.
4、The description of work in the article is relatively vague, failing to accurately portray your job responsibilities.
5、There is an issue with the data in Figure 11, please correct it.
6、I believe that there are many external influences on the image in a real scene, such as reflections, headlights, etc., so please suggest possible solutions or outlooks.
7、In the Contribution section, the fifth point is not reflected in the paper.
8、The paper presents some grammar mistakes, please proofread the whole paper and revise them.
9、I'm glad you mentioned some recent related work, including "Sequential Order-Aware Coding-Based Robust Subspace Clustering for Human Action Recognition in Untrimmed Videos", "GAN- Siamese Network for Cross-Domain Vehicle Re-identification in Intelligent Transport Systems", "Spatio-Temporal Feature Encoding for Traffic Accident Detection in VANET Environment", and "An Efficient and Secure Identity-Based Signature System for Underwater Green Transport System". In addition, it is recommended that you add citations for these works.
Comments on the Quality of English Language
The paper presents some grammar mistakes, please proofread the whole paper and revise them.
Author Response
Please see the attachment.

Reviewer 3 Report (New Reviewer)
Comments and Suggestions for Authors
Dear authors,
Hereafter some comments from my side:
- On page 2, Section 1 (lines 46-47), what do you mean when you wrote “However, the system’s observations and points need to be more precise”? It is not clear what you are referring to…
- Same page, towards the end, when you talk about the Manchester Driver Behavior Questionnaire (DBQ), is it the one you used to train the system? Because it is not clear how you create your ground-truth… moreover, this method is not objective, but it depends on the self-assessment of participants.
- In addition, are you sure that the references – and, in particular, this one (you indicated as [7]) are correct?
- Overall, you should re-consider the text, because there are too many repetitions, meaning that the same concept is repeated too many times in different ways. This makes the paper too long and, therefore, not easy to read.
- In the same way, maybe it is not necessary to have a paragraph 1.1, because it is the only one (cancel it, or add more paragraphs like that, e.g., 1.2, 1.3, etc.).
- On page 5, you mentioned different ML methods with respect DL, which have good values of accuracy (not necessarily worse that DL approach); why did you consider only this approach? Maybe some explanations/discussions here can be needed. This is particularly important in Tables 1 and 2 (e.g., here you could present some numbers, if available).
- On page 5, figure 1, it is not fully convincing that Image Processing is regarded as a technique to determine drowsiness … or better, it should be included in the Behavioral-based Techniques branch. Or maybe, I misunderstood this figure.
- On page 9, it is not clear why you included the paragraph before Chapter 3; which is the goal? To show the usefulness of IoT? Maybe not necessary.
- On Section 3, paragraph 3.1, I think that the first part is not necessary, because already written in other parts of the article.
- I think that figures 2 and 3 would need more details in their capture (the context is difficult to understand as they are).
- On page 11, lines 367-368, what do you mean with “… to lower the parameters by random inactivation to prevent overfitting …”? How?
- On page 13, references [42] and [43] maybe are not necessary, or you should collocate them in a different position.
- It is not clear how you used the PERCLOS: if it is used to determine the drowsiness – as a kind of ground-truth – why do you need to train a classifier? Otherwise, is it an input to the classifier?
- Overall, you should clarify which are the inputs your classifier use.
- Your results seem very interesting and promising, but they are not explained in a totally clear way for example, the accuracy of table 3 seems to be different from the one reported in other parts (see the confusion matrix, or table 4).
- In addition, how did you compare your method with the others?
- Finally, as next steps, have you considered also the application of your method in real world (at least with datasets of this type)?
I hope my comments can help to improve your work.
Author Response
Please see the attachment.

Reviewer 4 Report (New Reviewer)
Comments and Suggestions for Authors
The text discusses the importance of a detection system in assessing the state of a driver's eyes to determine fatigue levels. It emphasizes the need for real-time alerts to prevent accidents and introduces the use of Internet of Things (IoT) technology in driver action recognition. The focus is on a Deep Learning model, specifically a CNN-Long Short Term Memory network, for detecting driver sleepiness. The study compares various algorithms, including EM-CNN, VGG-16, GoogleNet, AlexNet, ResNet50, and CNN-LSTM, with the latter showing superior accuracy.
The article requires several adjustment before considering it for publication.
INTRODUCTION
The context is superficially defined; more specific references to the considerations made are needed. For instance no reference support the statement "The risk of sleepy driving is six times higher for people who work night, rotating, or double shifts than for other categories of workers". As an example, a work that could help to better frame the context is: "Nonis et al. Can ADAS Distract Driver’s Attention? An RGB-D Camera and Deep Learning-Based Analysis."
CONTRIBUTION OF THE PAPER
In this Section, scientific contribution should be highlighted. Collecting drowsy and drowsiness-related videos and image datasets from Kaggle and YouTube surely does not fit with the definition of "contribution".
RELATED WORKS
The first part (lines 137-172) must be rewritten. It is not clear and the text refers to references that have not been cited.
Remove comments such as that one at page 8, line 270.
PROPOSED METHODOLOGY
The methodology sounds interesting, but the Section must be rephrased to improve clarity and readiness.
EXPERIMENTAL RESULTS
Provide some example frames of the selected input data.
IMPLEMENTATION PROCESS AND RESULTS DISCUSSION
For the sake of clarity, provide all the implementation process in one Section and "Results and Discussion" in another Section.
Figure 11 shows an awful result in terms of False Positive (0.98), is it a mistake?
There are some results of previous revisions. I cannot compeltely understand which is the final version I have to review.
IMAGES
For the sake of readiness, provide images with a much higher resolution.
Comments on the Quality of English LanguageQuality of English language must be greatly improved.
Round 2
Reviewer 1 Report (Previous Reviewer 2)
Comments and Suggestions for Authors
it can be accepted
Reviewer 3 Report (New Reviewer)
Comments and Suggestions for Authors
Dear authors,
Thanks a lot for addressing all my comments. No further ones from my side (the only “weaker” part remains your explanations about the almost exclusive use of DL).
Reviewer 4 Report (New Reviewer)
Comments and Suggestions for Authors
The authors have properly addressed all my comments and suggestions. In my opinion their effort has now made the paper suitable for publication.
This manuscript is a resubmission of an earlier submission. The following is a list of the peer review reports and author responses from that submission.
Round 1
Reviewer 1 Report
Comments and Suggestions for Authors
This article presents a Driver Drowsiness Detection framework that utilizes U-Net for extracting facial regions from video clips and then employs CNN-LSTM for recognition, achieving optimal performance compared to other deep learning methods.
However, the paper fails to provide a detailed description of the methodology, such as the optimization objectives (setting of the loss function) and the overall framework. I believe that a complete and detailed description would better showcase the authors' work.
Furthermore, the article lacks a detailed introduction to the experimental dataset and experimental settings, and it doesn't include specific ablation experiments. Providing more detailed experimental descriptions and conducting specific ablation experiments are expected to increase the credibility of the paper's results.
Finally, there are several spelling errors in the paper, and I recommend the authors carefully review and make necessary improvements in this regard.
Comments on the Quality of English LanguageOverall, the paper is clearly written, but there are several spelling errors throughout. The writing could be enhanced to meet academic style. It is strongly recommended that authors thoroughly review the paper and improve these aspects. The presence of "NA %" in Table 1, and "dataset" in Table 5, for instance, are apparent errors that should be rectified in a revised version.
Reviewer 2 Report
Comments and Suggestions for Authors
This paper proposed a CNN-Long Sort Time Memory (LSTM) network-based Deep Learning model to detect driver drowsiness. The dataset is a tested set of different techniques namely EM-CNN, VGG-16, GoogleNet, Alexnet, ResNet50 and CNN-LSTM. Those algorithms have been used for classification purposes among which it is clear that CNN-LSTM gives better accuracy than any other deep learning algorithm. The model is fed video clips of limited duration and it discriminates the clip based on the series of movements that the driver shows in the video clip
1. The problem statement, motivation and main findings of the proposed model should be added in the abstract section. Remove the acronym in the keyword.
2. Provide the nomenclature table below the abstract.
3. Contribution should be highlighted in 3 to 4 points and don't describe it in a paragraph. Provide the contribution in step by step manner.
4. Focus on the related work section add recent 5 papers in the related work section and clearly explain about the research gap in the existing method.
5. Overall architecture of the proposed model should be included in the article.
6. Equation must be written in an equation editor and avoid the overlapping of the variables in the equations.
7. Improve the quality of all the figures. Provide the flow chart of the proposed model.
8. A detailed elaboration is not given for the merits of utilising U-Net for image-to-image mapping.
9. To broaden the scope of this paper, the authors should refer to some work such as: Federated Learning Based on CTC for Heterogeneous Internet of Things,Fusion of blockchain, IoT and artificial intelligence-A survey and I-Health: SDN-based fog architecture for IIoT applications in healthcare
9. The writing organization of this paper should be developed better and also, I suggest you carefully check the paper to correct the types, grammar errors and sentence structures
Comments on the Quality of English Languagenone
Reviewer 3 Report
Comments and Suggestions for Authors
1、It is necessary to briefly describe the background of the project in the summary section, but the background description should be reduced in the section describing the technology.
2、The logic of the article is unclear, and I don't understand the connection between the sentence 'Therefore, driving is...' and the previous text.
3、In the Related Works section, the sentence lacks citation. The author introduced some methods but did not analyze the advantages and disadvantages of the methods.
4、In the section describing Figure 4, there are highly overlapping contents.
5、I am deeply puzzled by the sentence 'In our case, the frames...' regarding Figure 6.
6、To make the reader clearer, please describe Figure 6, Algorithm 1, RIO, PERCLOS, and POM and add units to the horizontal axis of Figure 7.
7、There is a problem with. 015 in the Implementation Process and Results Discussion section.
8、It is better to discuss some recent related work and describe the differences between them and the recent proposed work to improve the quality of this paper, e.g., Sequential Order-Aware Coding-Based Robust Subspace Clustering for Human Action Recognition in Untrimmed Videos, GAN-Siamese Network for Cross-Domain Vehicle Re-identification in Intelligent Transport Systems, Spatio-Temporal Feature Encoding for Traffic Accident Detection in VANET Environment, An Efficient and Secure Identity-Based Signature System for Underwater Green Transport System.
9、The author did not review the article carefully, and there were many small mistakes, such as missing a period, % should be added in table 3 and made a mistake of 21 77. Similarly, in Table 5, the units should also be included in the Compression and Drowsiness Detection Time.
Comments on the Quality of English Language
There are some grammar errors and logical issues in the article.
Round 2
Reviewer 1 Report
Comments and Suggestions for Authors
Thanks for the authors' efforts in their response. However, my concerns are still retained after carefully reviewing the response.
Firstly, the details of the loss function or training setups are still missing in the revised paper, which is crucial for the training process. An exact description of the loss function and training details is expected to be shown.
Secondly, ablation studies and detailed descriptions of the experimental dataset can't be found. The credibility and reproducibility of this work can be improved with more detailed descriptions.
Comments on the Quality of English LanguageThere are still some errors, but they don't interfere with understanding.
Reviewer 2 Report
Comments and Suggestions for Authors
While the authors have adeptly addressed most comments, there is an opportunity to broaden the paper’s scope by referencing additional significant works in the field. Federated Learning Based on CTC for Heterogeneous Internet of Things,Fusion of blockchain, IoT and artificial intelligence-A survey and I-Health: SDN-based fog architecture for IIoT applications in healthcare
Comments on the Quality of English LanguageWhile the authors have adeptly addressed most comments, there is an opportunity to broaden the paper’s scope by referencing additional significant works in the field. Federated Learning Based on CTC for Heterogeneous Internet of Things,Fusion of blockchain, IoT and artificial intelligence-A survey and I-Health: SDN-based fog architecture for IIoT applications in healthcare
Round 3
Reviewer 1 Report
Comments and Suggestions for Authors
Thank you very much for the efforts made by the author in the discussion. Regarding the new round of replies, I have the following suggestions:
1. All the charts should be carefully examined and modified. Many of the images are very blurry. The "Dataset (Images" in Table 5 and the "Dataset" in Table 6 are likely errors that have not been modified consistently throughout. The formatting in Table 6 is messy. These should have been addressed in previous versions.
2. The writing, structure, grammar, and formatting should be carefully revised. Despite the author's claims of careful checking, there are still many errors throughout the text, such as "paycharm," the formatting in Table 6, obvious spelling errors, and capitalization errors.
3. Detailed explanations should be provided for the formulas and the use of symbols. Additionally, there are errors in many formula edits. In the description of the loss function, the text and the formula symbols do not match.
4. I noticed that the author mentioned the use of loss as an evaluation metric in line 691, but I did not see any corresponding experimental results. Furthermore, the loss function should not be used as the evaluation metric for different models. I suggest modifying the relevant description.
5. The article lacks detailed descriptions of evaluation metrics, such as sensitivity and specificity. The evaluation for segmentation and training are also missing, leading a blank on the effectiveness of the segmentation result.
Same as above.
Round 4
Reviewer 1 Report
Comments and Suggestions for Authors
During multiple discussions, the author did not address my suggestions to supplement the relevant experiments to demonstrate the effectiveness of this work. For example, there are no ablation studies or metrics about segmentation. The contributions of the work lack experimental support.
Additionally, the author did not adhere to the writing conventions of AI papers, making it difficult to comprehend the work.
There are severe issues with the figures in the paper. Figure 6 fails to clearly demonstrate the whole workflow, making the inference process unclear. Figure 7 appears to have directly copied someone else's illustration, which is unacceptable. The significance of Figures 9 and 10 is unclear. The usage of the term "category" in Table 6 is still inaccurate. It is recommended that the author seriously study academic writing and improve the article.
The evaluation metrics in the paper are confusing, as sensitivity and recall are treated as the same metric, but the author seems to have mixed their usage.
I suggest that the loss function should not be used as an evaluation metric, but the author has gone against this recommendation by including the loss function as an evaluation metric in Table 4.
In conclusion, despite multiple revisions, the article and work still have significant deficiencies and fail to meet the standards for publication. I suggest that the author should consolidate their work, reconsider the article's structure, and pay attention to the writing.
Comments on the Quality of English LanguageSee comments above.
